# Not All Tokens Are What You Need for Pretraining

**Zhenghao Lin**[⋆χϕ] **Zhibin Gou**[⋆πϕ] **Yeyun Gong**[◇ϕ] **Xiao Liu**[ϕ] **Yelong Shen**[ϕ]
**Ruochen Xu**[ϕ] **Chen Lin**[◇χρ] **Yujiu Yang**[◇π] **Jian Jiao**[ϕ] **Nan Duan**[ϕ] **Weizhu Chen**[ϕ]

[χ]Xiamen University   [π]Tsinghua University   [ρ]Shanghai AI Laboratory
[ϕ]Microsoft
https://aka.ms/rho

## Abstract

Previous language model pre-training methods have uniformly applied a next-token prediction loss to all training tokens. Challenging this norm, we posit that *"Not all tokens in a corpus are equally important for language model training"*. Our initial analysis examines token-level training dynamics of language model, revealing distinct loss patterns for different tokens. Leveraging these insights, we introduce a new language model called RHO-1. Unlike traditional LMs that learn to predict every next token in a corpus, RHO-1 employs Selective Language Modeling (SLM), which selectively trains on useful tokens that aligned with the desired distribution. This approach involves scoring tokens using a reference model, and then training the language model with a focused loss on tokens with higher scores. When continual pretraining on 15B OpenWebMath corpus, RHO-1 yields an absolute improvement in few-shot accuracy of up to 30% in 9 math tasks. After fine-tuning, RHO-1-1B and 7B achieved state-of-the-art results of 40.6% and 51.8% on MATH dataset, respectively — matching DeepSeekMath with only 3% of the pretraining tokens. Furthermore, when continual pretraining on 80B general tokens, RHO-1 achieves 6.8% average enhancement across 15 diverse tasks, increasing both data efficiency and performance of the language model pre-training.

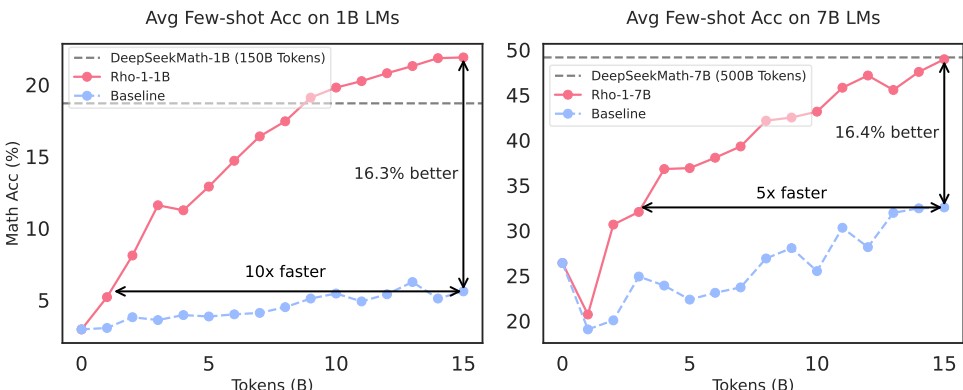

Figure 1: We continual pretrain 1B and 7B LMs with 15B OpenWebMath tokens. RHO-1 is trained with our proposed Selective Language Modeling (SLM), while baselines are trained using causal language modeling. SLM improves average few-shot accuracy on GSM8k and MATH by over 16%, achieving the baseline performance 5-10x faster.

---

[⋆]Equal contribution. See author contributions for details. Work done during their internships at Microsoft Research Asia. ✉: zhenghaolin@stu.xmu.edu.cn; zebgou@gmail.com
[◇]Correspondence authors.

38th Conference on Neural Information Processing Systems (NeurIPS 2024).

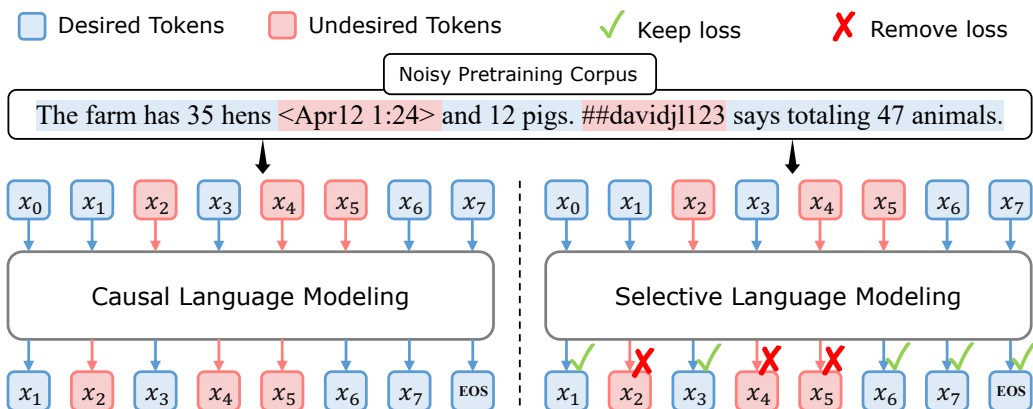

Figure 2: **Upper:** Even an extensively filtered pretraining corpus contains token-level noise. **Left:** Previous Causal Language Modeling (CLM) trains on all tokens. **Right:** Our proposed Selective Language Modeling (SLM) selectively applies loss on those useful and clean tokens.

# 1 Introduction

Scaling up model parameters and dataset size has consistently elevated the next-token prediction accuracy in large language models, yielding significant advancements in artificial intelligence [Kaplan et al., 2020, Brown et al., 2020, OpenAI, 2023, Team et al., 2023]. However, training on all available data is not always optimal or feasible. As a result, the practice of data filtering has become crucial, using various heuristics and classifiers [Brown et al., 2020, Wenzek et al., 2019] to select training documents. These techniques significantly improve data quality and boost model performance.

However, despite thorough document-level filtering, high-quality datasets still contain many noisy tokens that can negatively affect training, as illustrated in Figure 2 (Upper). Removing such tokens might alter the text's meaning, while overly strict filtering could exclude useful data [Welbl et al., 2021, Muennighoff et al., 2024] and lead to biases [Dodge et al., 2021, Longpre et al., 2023]. Furthermore, research indicates that the distribution of web data does not inherently align with the ideal distribution for downstream applications [Tay et al., 2022, Wettig et al., 2023]. For example, common corpus at the token level may include undesirable content like hallucinations or highly ambiguous tokens that are hard to predict. Applying the same loss to all tokens can lead to inefficient computation on non-essential tokens, potentially restricting LLMs from achieving more advanced levels of intelligence.

To explore how language models learn at the token level, we initially examined training dynamics, particularly how the token-level loss evolves during usual pretraining. In §2.1, we evaluated the model's token perplexity at different checkpoints and categorized tokens into different types. Our findings reveal that significant loss reduction is limited to a select group of tokens. Many tokens are "easy tokens" that are already learned, and some are "hard tokens" that exhibit variable losses and resist convergence. These tokens can lead to numerous ineffective gradient updates.

Based on these analyses, we introduce RHO-1 models trained with a novel Selective Language Modeling (SLM) objective [3]. As shown in Figure 2 (Right), this approach inputs the full sequence into the model and selectively removes the loss of undesired tokens. The detailed pipeline is depicted in Figure 4: First, SLM trains a reference language model on high-quality corpora. This model establishes utility metrics to score tokens according to the desired distribution, naturally filtering out unclean and irrelevant tokens. Second, SLM uses the reference model to score each token in a corpus using its loss (§2.2). Finally, we train a language model only on those tokens that exhibit a high excess loss between the reference and the training model, selectively learning the tokens that best benefit downstream applications (§2.2).

We show through comprehensive experiments that SLM significantly enhances token efficiency during training and improves performance on downstream tasks. Furthermore, our findings indicate that SLM effectively identifies tokens relevant to the target distribution, resulting in improved perplexity scores

---

[3]"Rho" denotes selective modeling of tokens with higher information "density ($\rho$)".

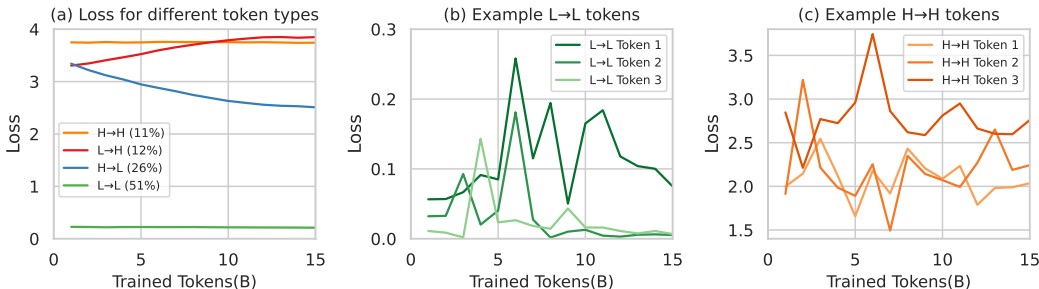

Figure 3: **The loss of four categories of tokens during pretraining.** (a) shows the loss of H→H, L→H, H→L, and L→L tokens during pretraining. (b) and (c) show three cases of fluctuating tokens' loss in L→L and H→H during pretraining, respectively.

on benchmarks for models trained with the selected tokens. §3.2 shows the effectiveness of SLM on math continual pretraining: both 1B and 7B RHO-1 outperform CLM-trained baselines by over 16% on the GSM8k and MATH datasets. SLM reaches baseline accuracy up to 10x faster, as shown in Figure 1. Remarkably, RHO-1-7B matches the state-of-the-art performance of DeepSeekMath-7B using only 15B tokens, compared to the 500B tokens required by DeepSeekMath. Upon fine-tuning, RHO-1-1B and 7B achieve 40.6% and 51.8% on MATH, respectively. Notably, RHO-1-1B is the first 1B LM to exceed 40% accuracy, nearing the early GPT-4's CoT performance of 42.5%. §3.3 confirms the efficacy of SLM in general continual pretraining: Training Tinyllama-1B on 80B tokens with SLM improves 6.8% on average across 15 benchmarks, with gains over 10% in code and math tasks. In §3.4, we demonstrate that in settings without high-quality reference data, we can use SLM for self-referencing, leading to an average improvement of up to 3.3% in downstream tasks.

## 2 Selective Language Modeling

### 2.1 Not All Tokens Are Equal: Training Dynamics of Token Loss

Our investigation begins with a critical look at how individual tokens' losses evolve during standard pre-training. We continue pre-training Tinyllama-1B with 15B tokens from OpenWebMath, saving checkpoints after every 1B tokens. We then evaluate token-level loss at these intervals using the validation set of approximately 320,000 tokens. Figure 3(a) reveals a striking pattern: tokens fall into four categories based on their loss trajectory—persistent high loss (H→H), increasing loss (L→H), decreasing loss (H→L), and consistent low loss (L→L). For further details on these categories, see §D.1. Our analysis uncovers that a mere 26% of tokens show a notable loss reduction (H→L), while the majority (51%) remain in the L→L category, indicating they have already been learned. Interestingly, 11% of the tokens are persistently challenging (H→H), likely due to high aleatoric uncertainty [Hüllermeier and Waegeman, 2021]. Additionally, 12% of tokens experience an unexpected loss increase (L→H) during training.

Our second observation is that a significant number of token losses exhibit persistent fluctuations, and resist convergence. The loss of many L→L and H→H tokens, as depicted in Figure 3 (b) and (c), show high variance during training. In §D.2, we visualize and analyze the content of these tokens and find that many of them are noisy, which is consistent with our hypothesis.

Consequently, we learn that the loss associated with each token during training does not decrease smoothly like the overall loss; instead, there is a complex training dynamic among different tokens. If we can select the appropriate tokens for the model to focus on during training, we may be able to stabilize the trajectory of the model's training and enhance its data efficiency.

### 2.2 Selective Language Modeling

**Overview**   Inspired by the practice of reference model in document-level filtering, we propose a simple pipeline of token-level data selection, termed "Selective Language Modeling (SLM)". Our method comprises three steps, as depicted in Figure 4. We begin by training a reference model on a curated, high-quality dataset. This model then assesses the loss of each token within the pretraining

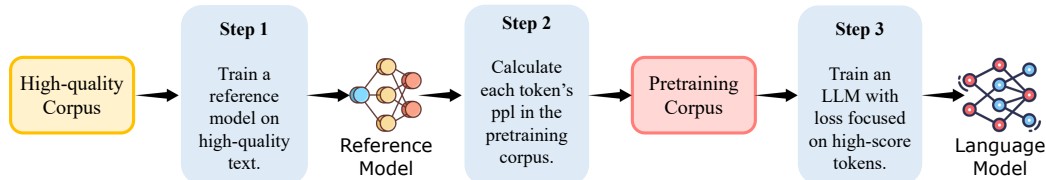

Figure 4: **The pipeline of Selective Language Modeling (SLM).** SLM optimizes language model performance by concentrating on valuable, clean tokens during pre-training. It involves three steps: (Step 1) Initially, train a reference model on high-quality data. (Step 2) Then, score each token's loss in a corpus using the reference model. (Step 3) Finally, selectively train the language model on tokens that have higher scores.

corpus. In the final phase, we train the language model selectively, focusing on tokens with high excess loss between the training and reference model. The intuition is that tokens with high excess loss are more learnable and better aligned with the desired distribution, naturally excluding tokens that are either irrelevant or of low quality. Below, we provide a detailed description of each step.

**Reference Modeling**   We begin by curating a high-quality dataset that reflects the desired data distribution. We train a reference model (RM) using standard cross-entropy loss on the curated data. The resulting RM is then used to assess the token loss within a larger pretraining corpus. We compute the reference loss ($\mathcal{L}_{\text{RM}}$) of a token $x_i$ based on the probability that the RM assigns to this token. The calculation is formalized as follows:

$$\mathcal{L}_{\text{RM}}(x_i) = -\log P(x_i|x_{<i}) \tag{1}$$

By evaluating $\mathcal{L}_{\text{RM}}$ for each token, we establish the reference loss for selective pretraining, allowing us to focus on the most influential tokens in language modeling.

**Selective Pretraining**   Note that Causal Language Modeling (CLM) employs the cross-entropy loss:

$$\mathcal{L}_{\text{CLM}}(\theta) = -\frac{1}{N}\sum_{i=1}^{N}\log P(x_i|x_{<i};\theta) \tag{2}$$

Here, $\mathcal{L}_{\text{CLM}}(\theta)$ represents the loss function parameterized by model $\theta$. $N$ is the length of the sequence, $x_i$ is the $i$-th token in the sequence, and $x_{<i}$ represents all tokens before the $i$-th token. In contrast, Selective Language Modeling (SLM) trains the language model with a focus on tokens that exhibit a high excess loss when compared to the reference model. The excess loss ($\mathcal{L}_\Delta$) for a token $x_i$ is defined as the difference between the current training model loss ($\mathcal{L}_\theta$) and the reference loss:

$$\mathcal{L}_\Delta(x_i) = \mathcal{L}_\theta(x_i) - \mathcal{L}_{\text{RM}}(x_i) \tag{3}$$

We introduce a token selection ratio $k\%$, which determines the proportion of tokens to be included based on their excess loss. The cross-entropy loss for the selected tokens is computed as follows:

$$\mathcal{L}_{\text{SLM}}(\theta) = -\frac{1}{N * k\%}\sum_{i=1}^{N} I_{k\%}(x_i) \cdot \log P(x_i|x_{<i};\theta) \tag{4}$$

Here, $N * k\%$ defines the number of tokens that fall within the top $k\%$ of excess loss. The indicator function $I_{k\%}(x_i)$ is defined as:

$$I_{k\%}(x_i) = \begin{cases} 1 & \text{if } x_i \text{ ranks in the top } k\% \text{ by } S(x_i) \\ 0 & \text{otherwise} \end{cases} \tag{5}$$

Table 1: **Few-shot CoT reasoning results of math pretraining.** All models are tested with few-shot prompting. Previous best results are highlighted in blue, while our best results are in purple. *Only unique math-related tokens are calculated. For RHO-1, we calculate only the selected tokens that are used for training. †We use OpenAI's MATH subset [Lightman et al., 2023] for evaluation, since some original test samples have been used in public training sets such as PRM800k. ‡The SAT only has 32 four-choice problems, so we average our results over the last three checkpoints, if available.

| Model | $\|\theta\|$ | Data | Uniq. Toks* | Train Toks | GSM8K | MATH† | SVAMP | ASDiv | MAWPS | TAB | MQA | MMLU STEM | SAT‡ | AVG |
|---|---|---|---|---|---|---|---|---|---|---|---|---|---|---|
| | | | | | 1-2B Base Models | | | | | | | | | |
| Tinyllama | 1.1B | - | - | - | 2.9 | 3.2 | 11.0 | 18.1 | 20.4 | 12.5 | 14.6 | 16.1 | 21.9 | 13.4 |
| Phi-1.5 | 1.3B | - | - | - | 32.4 | 4.2 | 43.4 | 53.1 | 66.2 | 24.4 | 14.3 | 21.8 | 18.8 | 31.0 |
| Qwen1.5 | 1.8B | - | - | - | 36.1 | 6.8 | 48.5 | 63.6 | 79.0 | 29.2 | 25.1 | 31.3 | 40.6 | 40.0 |
| Gemma | 2.0B | - | - | - | 18.8 | 11.4 | 38.0 | 56.6 | 72.5 | 36.9 | 26.8 | 34.4 | 50.0 | 38.4 |
| DeepSeekLLM | 1.3B | OWM | 14B | 150B | 11.5 | 8.9 | - | - | - | - | - | 29.6 | 31.3 | - |
| DeepSeekMath | 1.3B | - | 120B | 150B | 23.8 | 13.6 | - | - | - | - | - | 33.1 | 56.3 | - |
| | | | | | Continual Pretraining on Tinyllama-1B | | | | | | | | | |
| Tinyllama-CT | 1.1B | OWM | 14B | 15B | 6.4 | 2.4 | 21.7 | 36.7 | 47.7 | 17.9 | 13.9 | 23.0 | 25.0 | 21.6 |
| RHO-1-Math | 1.1B | OWM | 14B | 9B | 29.8 | 14.0 | 49.2 | 61.4 | 79.8 | 25.8 | 30.4 | 24.7 | 28.1 | 38.1 |
| Δ | | | | -40% | +23.4 | +11.6 | +27.5 | +24.7 | +32.1 | +7.9 | +16.5 | +1.7 | +3.1 | **+16.5** |
| RHO-1-Math | 1.1B | OWM | 14B | 30B | 36.2 | 15.6 | 52.1 | 67.0 | 83.9 | 29.0 | 32.5 | 23.3 | 28.1 | 40.9 |
| | | | | | $\geq$ 7B Base Models | | | | | | | | | |
| LLaMA-2 | 7B | | - | - | 14.0 | 3.6 | 39.5 | 51.7 | 63.5 | 30.9 | 12.4 | 32.7 | 34.4 | 31.4 |
| Mistral | 7B | | - | - | 41.2 | 11.6 | 64.7 | 68.5 | 87.5 | 52.9 | 33.0 | 49.5 | 59.4 | 52.0 |
| Minerva | 8B | - | 39B | 164B | 16.2 | 14.1 | - | - | - | - | - | 35.6 | - | - |
| Minerva | 62B | - | 39B | 109B | 52.4 | 27.6 | - | - | - | - | - | 53.9 | - | - |
| Minerva | 540B | - | 39B | 26B | 58.8 | 33.6 | - | - | - | - | - | 63.9 | - | - |
| LLemma | 7B | PPile | 55B | 200B | 38.8 | 17.2 | 56.1 | 69.1 | 82.4 | 48.7 | 41.0 | 45.4 | 59.4 | 50.9 |
| LLemma | 34B | PPile | 55B | 50B | 54.2 | 23.0 | 67.9 | 75.7 | 90.1 | 57.0 | 49.8 | 54.7 | 68.8 | 60.1 |
| Intern-Math | 7B | - | 31B | 125B | 41.8 | 14.4 | 61.6 | 66.8 | 83.7 | 50.0 | 57.3 | 24.8 | 37.5 | 48.7 |
| Intern-Math | 20B | - | 31B | 125B | 65.4 | 30.0 | 75.7 | 79.3 | 94.0 | 50.9 | 38.5 | 53.1 | 71.9 | 62.1 |
| DeepSeekMath | 7B | - | 120B | 500B | 64.1 | 34.2 | 74.0 | 83.9 | 92.4 | 63.4 | 62.4 | 56.4 | 84.4 | 68.4 |
| | | | | | Continual Pretraining on Mistral-7B | | | | | | | | | |
| Mistral-CT | 7B | OWM | 14B | 15B | 42.9 | 22.2 | 68.6 | 71.0 | 86.1 | 45.1 | 47.7 | 52.6 | 65.6 | 55.8 |
| RHO-1-Math | 7B | OWM | 14B | 10.5B | 66.9 | 31.0 | 77.8 | 79.0 | 93.9 | 49.9 | 58.7 | 54.6 | 84.4 | 66.2 |
| Δ | | | | -30% | +24.0 | +8.8 | +9.2 | +8.0 | +7.8 | +4.8 | +11.0 | +2.0 | +18.8 | **+10.4** |

By default, we use $\mathcal{L}_\Delta$ as the score function $S$. This ensures that the loss is applied only to the tokens that are deemed most beneficial for the language model to learn from. In practice, token selection can be implemented by ranking the tokens in a batch according to their excess loss and using only the top $k\%$ of tokens for training. This process eliminates the loss for undesired tokens without incurring additional costs during pretraining, making our approach both efficient and easily integrated.

## 3 Experiments

We continually pretrained models in both mathematical and general domain and designed ablation and analysis experiments to understand the effectiveness of SLM.

### 3.1 Experimental Setup

**Reference Model Training** To train our mathematical reference model, we gathered a dataset of 0.5B high-quality, math-related tokens. This dataset is a blend of synthetic data from GPT [Yu et al., 2024, Huang et al., 2024] and manually curated data [Yue et al., 2024, Ni et al., 2024]. For the general reference model, we compiled a corpus of 1.9B tokens from open-source datasets, such as Tulu-v2 [Ivison et al., 2023] and OpenHermes-2.5 [Teknium, 2023]. We trained the reference models for 3 epochs. The maximum learning rate was set at 5e-5 for 1B models and 1e-5 for 7B models, applying a cosine decay schedule. We set the maximum sequence lengths to 2048 for 1B models and 4096 for 7B models, packing multiple samples into these lengths for model input. In all main experiments, we initialized the continual pretraining model and the reference model with the *same* base model.

Table 2: **Tool-integrated reasoning results of math pretraining.**

| Model | Size | Tools | SFT Data | GSM8k | MATH | SVAMP | ASDiv | MAWPS | TAB | GSM-H | AVG |
|---|---|---|---|---|---|---|---|---|---|---|---|
| **Used for SFT?** | | | | ✓ | ✓ | ✗ | ✗ | ✗ | ✗ | ✗ | |
| *Previous Models* | | | | | | | | | | | |
| GPT4-0314 | - | ✗ | - | 92.0 | 42.5 | 93.1 | 91.3 | 97.6 | 67.1 | 64.7 | 78.3 |
| GPT4-0314 (PAL) | - | ✓ | - | 94.2 | 51.8 | 94.8 | 92.6 | 97.7 | 95.9 | 77.6 | 86.4 |
| MAmmoTH | 70B | ✓ | MI-260k | 76.9 | 41.8 | 82.4 | - | - | - | - | - |
| ToRA | 7B | ✓ | ToRA-69k | 68.8 | 40.1 | 68.2 | 73.9 | 88.8 | 42.4 | 54.6 | 62.4 |
| ToRA | 70B | ✓ | ToRA-69k | 84.3 | 49.7 | 82.7 | 86.8 | 93.8 | 74.0 | 67.2 | 76.9 |
| DeepSeekMath | 7B | ✓ | ToRA-69k | 79.8 | 52.0 | 80.1 | 87.1 | 93.8 | 85.8 | 63.1 | 77.4 |
| *Our Pretrained Models* | | | | | | | | | | | |
| TinyLlama-CT | 1B | ✓ | ToRA-69k | 51.4 | 38.4 | 53.4 | 66.7 | 81.7 | 20.5 | 42.8 | 50.7 |
| RHO-1-Math | 1B | ✓ | ToRA-69k | 59.4 | 40.6 | 60.7 | 74.2 | 88.6 | 26.7 | 48.1 | 56.9 |
| Δ | | | | +8.0 | +2.2 | +7.3 | +7.5 | +6.9 | +6.2 | +5.3 | **+6.2** |
| Mistral-CT | 7B | ✓ | ToRA-69k | 77.5 | 48.4 | 76.9 | 83.8 | 93.4 | 67.5 | 60.4 | 72.6 |
| RHO-1-Math | 7B | ✓ | ToRA-69k | 81.3 | 51.8 | 80.8 | 85.5 | 94.5 | 70.1 | 63.1 | 75.3 |
| Δ | | | | +3.8 | +3.4 | +3.9 | +1.7 | +1.1 | +2.6 | +2.7 | **+2.7** |

**Pretraining Corpus**    For mathematical reasoning, we utilize the OpenWebMath (OWM) dataset [Paster et al., 2023], which comprises approximately 14B tokens sourced from math-related web pages in the Common Crawl. In the general domain, we combine the SlimPajama [Daria et al., 2023] and StarCoderData [Li et al., 2023a] (both part of the Tinyllama corpus) with OpenWebMath, training on a total of 80 billion tokens with a mix ratio of 6:3:1.

**Pretraining Setting**    For math pretraining, we continue pretraining on the Tinyllama-1.1B model [Zhang et al., 2024] and the Mistral-7B model [Jiang et al., 2023] with learning rates of 8e-5 and 2e-5, respectively. For the 1.1B model, we conducted our training on 32 × H100 80G GPUs. This configuration allowed us to train approximately 15 billion tokens in around 3.5 hours and 50 billion tokens in about 12 hours. In the case of the 7B model, training the same 15 billion tokens took approximately 18 hours under similar hardware conditions. For general domain, we set the learning rate for Tinyllama-1.1B model to 1e-4 and train 80B tokens under the same hardware conditions, which takes approximately 19 hours. The batch size is uniformly set to 1M tokens for both domains. Regarding the token selection ratio, we use 60% for the Tinyllama-1.1B model and 70% for the Mistral-7B model.

**Baseline Setting**    We use models that have been continually pretrained (Tinyllama-CT and Mistral-CT) through regular causal language modeling as baselines. Moreover, we compare RHO-1 with well-known and top-performing baselines, including Gemma [Team et al., 2024], Qwen1.5 [Bai et al., 2023], Phi-1.5 [Li et al., 2023b], DeepSeekLLM [DeepSeek-AI, 2024], DeepSeekMath [Shao et al., 2024], CodeLlama [Roziere et al., 2023], Mistral [Jiang et al., 2023], Minerva [Lewkowycz et al., 2022], Tinyllama [Zhang et al., 2024], LLemma [Azerbayev et al., 2023], and InternLM2-Math [Ying et al., 2024]. For fine-tuning results, we also compare with previous best models MAmmoTH[Yue et al., 2024] and ToRA[Gou et al., 2024].

**Evaluation Setup**    To comprehensively evaluate pretrained models, we compare their few-shot capabilities and fine-tuning performance across a variety of tasks. We adopt the lm-eval-harness[4] [Gao et al., 2023] for general tasks, and develop math evaluation suite[5] for math tasks. We use vllm (v0.3.2) [Kwon et al., 2023] to speed up inference. Further details on our evaluation can be found in Appendix E.

## 3.2    Math Pre-training Results

**Few-shot CoT Reasoning Results**    We evelate base models prompting with few-shot chain-of-thought (CoT) [Wei et al., 2022a] examples following previous works [Lewkowycz et al., 2022, Azerbayev et al., 2023, Shao et al., 2024]. As results shown in Table 1, in comparison to continue pretraining directly, RHO-1-Math has achieved the average few-shot accuracy improvement of

---

[4]https://github.com/EleutherAI/lm-evaluation-harness
[5]https://github.com/ZubinGou/math-evaluation-harness

16.5% on 1B models and 10.4% on 7B models. Furthermore, after training for multiple epochs on OpenWebMath, we find that RHO-1 could further increase the average few-shot accuracy to 40.9%. Compared to DeepSeekMath-7B, which pretrained on 500 billion math-related tokens, RHO-1-7B pretrained on only 15 billion tokens (selecting 10.5 billion tokens) achieved comparable results, demonstrating the efficiency of our approach.

**Tool-Integrated Reasoning Results**    We fine-tune RHO-1 and baseline models on 69k ToRA corpus [Gou et al., 2024], consisting of 16k GPT-4-generated trajectories in a tool-integrated reasoning format, and 53k answer-augmented samples using LLaMA. As presented in Table 2, RHO-1-1B and RHO-1-7B achieved a state-of-the-art 40.6% and 51.8% on MATH dataset, respectively. On some unseen tasks (*e.g.,* TabMWP and GSM-Hard), RHO-1 also demonstrates a certain degree of generalizability, with an average few-shot accuracy improvement of 6.2% on the RHO-1-Math-1B and 2.7% on RHO-1-Math-7B.

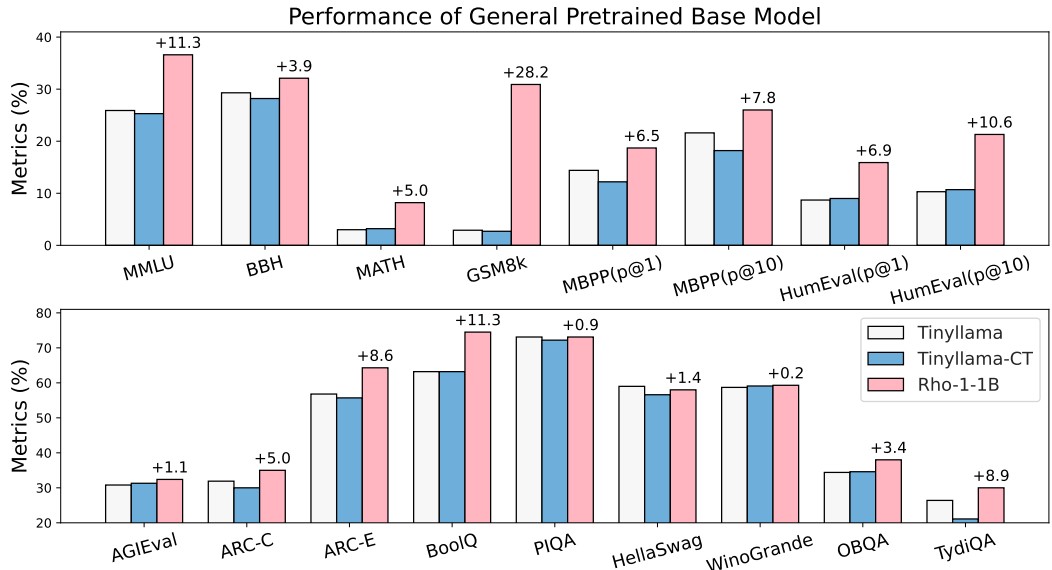

Figure 5: **General pretraining results.** We continual pretraining Tinyllama-1B on 80G general tokens. Tinyllama-CT is etrained with CLM, while RHO-1 is trained with our proposed SLM.

## 3.3    General Pre-training Results

We confirm the efficacy of the SLM in general pretraining by continual training Tinyllama-1.1B on 80 billion tokens. The results depicted in Figure 5 indicate that although Tinyllama has already undergone extensive training on the majority of these tokens, the application of SLM yields an average enhancement of 6.8% across 15 benchmarks compared to direct continual pretraining. The improvements were especially pronounced in code and math tasks, exceeding 10%.

## 3.4    Self-Reference Results

In this section, we demonstrate that SLM can enhance the effectiveness of model pre-training using only pre-training corpora, without the need for additional high-quality data. Specifically, we initially trained the reference model on the OpenWebMath (OWM) corpus, a subset of Proof-Pile-2 (PPile). We evaluated OWM and PPile using the trained reference model and selected tokens for training. In this scenario, we assume the absence of downstream task-related data, a common situation in real-world applications. We hypothesize that the key factor is not scoring the desired distribution but filtering out noisy tokens. Therefore, we employed two different scoring functions based on the reference model loss, $\mathcal{L}_{RM}$, and the information entropy of the next token, $\mathcal{H}_{RM}$, which measures the uncertainty of the next token. Details are provided in Appendix H.

Table 3: **Self-Reference results.** We use OpenWebMath (OWM) to train the reference model.

| Model | Score Function | Data | Uniq. Toks | Train Toks | GSM8K | MATH | SVAMP | ASDiv | MAWPS | MQA | AVG |
|---|---|---|---|---|---|---|---|---|---|---|---|
| Tinyllama-CT (RM) | - | OWM | 14B | 15B | 6.3 | 2.6 | 21.7 | 36.7 | 47.7 | 13.9 | 21.5 |
| Tinyllama-SLM | $\mathcal{L}_{\mathrm{RM}}$ | OWM | 14B | 10.5B | 6.7 | 4.6 | 23.3 | 40.0 | 54.5 | 14.3 | 23.9 |
| Tinyllama-SLM | $\mathcal{H}_{\mathrm{RM}}$ | OWM | 14B | 10.5B | 7.0 | 4.8 | 23.0 | 39.3 | 50.5 | 13.5 | 23.0 |
| Tinyllama-SLM | $\mathcal{L}_{\mathrm{RM}} \cap \mathcal{H}_{\mathrm{RM}}$ | OWM | 14B | 9B | 7.1 | 5.0 | 23.5 | 41.2 | 53.8 | 18.0 | 24.8 |
| Tinyllama-CT | - | PPile | 55B | 52B | 8.0 | 6.6 | 23.8 | 41.0 | 54.7 | 14.2 | 24.7 |
| Tinyllama-SLM | $\mathcal{L}_{\mathrm{RM}} \cap \mathcal{H}_{\mathrm{RM}}$ | PPile | 55B | 36B | 8.6 | 8.4 | 24.4 | 43.6 | 57.9 | 16.1 | 26.5 |

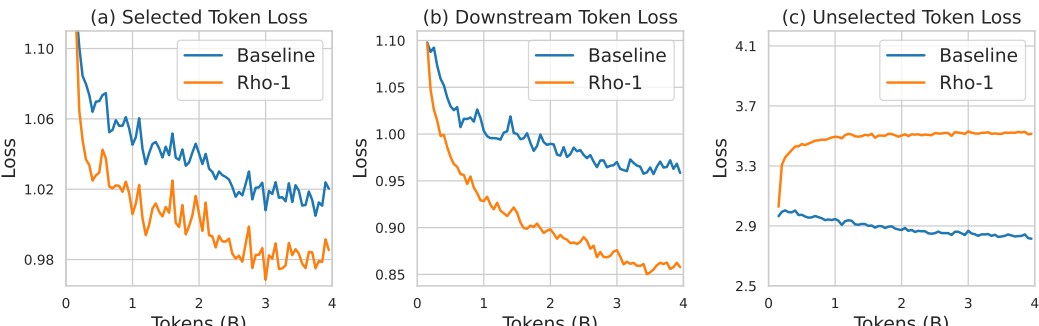

Figure 6: **The dynamics of pretraining loss and downstream loss.** (a) and (c) represent the loss of tokens selected/unselected by SLM during pretraining in both SLM and CLM methods, while (b) represents the loss of the SLM and CLM methods on MetaMath [Yu et al., 2024]. We tested the above results through the process of pretraining with a total of 4 billion tokens.

The experimental results, as shown in Table 3, indicate that using only the OWM-trained reference model can effectively guide the model in pre-training on the same corpus, improving average downstream performance by +2.4%. Using only the information entropy as the score function brought about a similar improvement. Additionally, we considered training on the intersection of tokens selected by the two scoring functions and found better performance, with a 40% reduction in tokens and +3.3% performance. Furthermore, training the SLM on the PPile, despite only using the OWM subset to train the reference model, still achieved a 1.8% improvement with 30% fewer tokens used. For more details, please refer to Appendix H.

## 3.5 Ablation Study and Analysis

**Selected Token Loss Aligns Better with Downstream Performance**   We utilized the reference model to filter tokens and assess their impact on validation and downstream losses after training. As depicted in Figure 6, we pretrained on 4B tokens and tracked loss variations across methods and validation sets. The RHO-1 showed greater loss reduction on selected tokens than regular pretraining. Cross-referencing figures (a), (b), and (c) reveals that selected-token pretraining substantially lowers downstream loss, while traditional pretraining's effect on downstream loss is less pronounced despite initial loss reductions. Therefore, we expect that selecting tokens for pretraining is more efficient.

In Figure 7, we demonstrate that the loss of selected tokens correlates with downstream task performance, following a power law similar to recent findings [Gadre et al., 2024]. Our analysis shows that tokens selected by SLM positively impact performance, while those not selected have a negative impact. Thus, reducing loss across all tokens is not imperative for improved model performance. Refer to Appendix F for further details.

**What Tokens are Selected with SLM?**   We aim to analyze the tokens selected by the SLM method in pretraining to further explore its working mechanism. To this end, we visualize the token selection process during the training of RHO-1 using the OpenWebMath. In §G.1, we have highlighted in blue the tokens that were retained during actual pretraining. We observe that the majority of tokens chosen

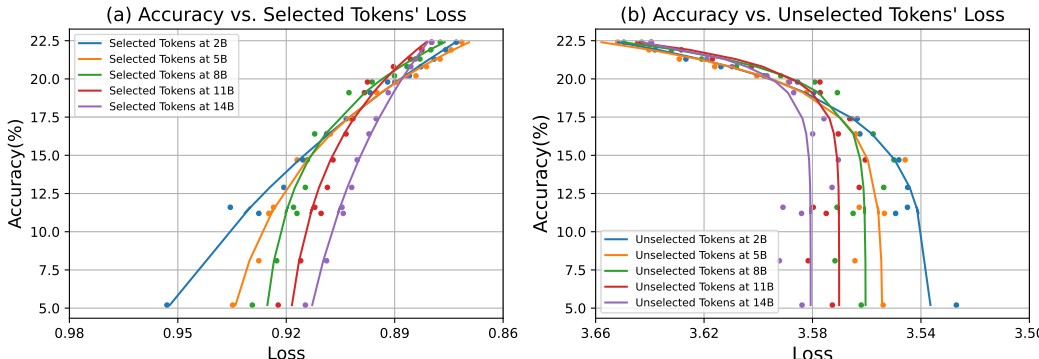

Figure 7: **The relationship between the selected tokens / unselected tokens loss in SLM and downstream task performance.** The y-axis represents the average few-shot accuracy on GSM8k and MATH. The x-axis represents the average loss on selected tokens / unselected tokens at corresponding checkpoint (2B, 5B, 8B, 11B, and 14B).

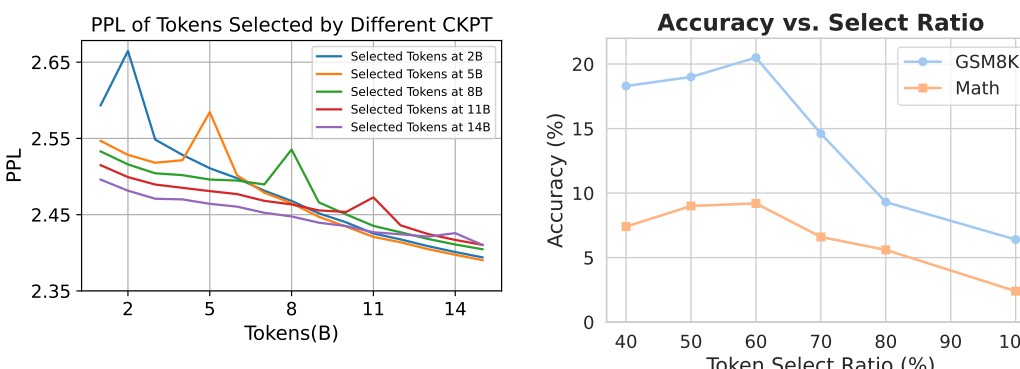

Figure 8: **The PPL of tokens selected by different checkpoint.** We test the PPL of the tokens selected at 2B, 5B, 8B, 11B, and 14B.

Figure 9: **Effect of token select ratio.** We train 1B LM with SLM objective on 5B tokens.

by the SLM method are closely related to mathematics, effectively training the model on the parts of the original corpus that are pertinent to mathematical content.

Furthermore, we investigated the differences in token filtering across various checkpoints during the training process and tested the perplexity of these tokens on different checkpoints. As illustrated in Figure 8, we found that the tokens selected by later checkpoints tend to have higher perplexity towards the later stages of training and lower perplexity in the earlier stages. This may suggest that the model first optimizes tokens with a larger learnable space, thereby increasing learning efficiency. Moreover, we noticed a sample-wise "double descent" [Nakkiran et al., 2021] on the loss of selected tokens, where the select token's perplexity initially increases before decreases. This might be an effect of selecting tokens based on excess loss, targeting those most in need at each checkpoint.

**Effect of Token Select Ratio** We investigate the impact of token selecting ratios of the SLM. Generally, the selecting ratio is defined by heuristic rules, similar to the approach previously employed in the training of Masked Language Models (MLMs) [Devlin et al., 2019, Liu et al., 2019]. As shown in Figure 9, the selected tokens is suitable for accounting for about 60% of the original tokens.

## 4 Conclusion

In this paper, we propose using Selective Language Modeling(SLM) to train RHO-1, which select more suitable tokens for current pretraining stage. We conducted the detailed analysis of the loss of tokens during the pretraining process and found that not all tokens are equal during pretraining. Our

experiments and analysis in the fields of mathematics and general have demonstrated the effectiveness of the SLM method, emphasizing the importance of token level in the LLM pretraining process. In the future, how to improve pretraining of LLMs from the perspective of token level worthy of in-depth research.

### Acknowledgments

Zhenghao Lin and Chen Lin were supported by National Key R&D Program of China (No. 2022ZD0160501), the Natural Science Foundation of China (No.62372390,62432011). Zhibin Gou and Yujiu Yang were supported by the Shenzhen Science and Technology Program (JCYJ20220818101001004) and the "Graph Neural Network Project" of Ping An Technology (Shenzhen) Co., Ltd.

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

# Appendix

## Contents

# A  Author Contributions

Zhenghao Lin designed and implemented detailed token selection process, conducted extensive preliminary experiments, developed the pre-training and evaluation pipeline, conducted most of the pre-training experiments and analysis, implemented baselines, and significantly contributed to the writing. Zhibin Gou presented a preliminary proposal, introduced the method of using excess loss for reweighting tokens, compiled high-quality corpora, trained reference models, set up the fine-tuning and evaluation pipelines, designed the experimental analysis, and significantly contributed to the writing. Yeyun Gong proposed the initial project and co-led the project with Weizhu Chen, they offered extensive advice and guidance on experiments and writing, and oversaw team collaboration and resource management. Xiao Liu, Yelong Shen, Ruochen Xu, Chen Lin, Yujiu Yang, Jian Jiao, and Nan Duan offered research mentorship, coordinated the project, and contributed to the writing.

# B  Related Works

## B.1  Pretraining Data Optimization

The objective of optimizing pre-training corpora is to maximize the performance and efficiency of language model training by improving the quality and scale of the pretrain data mixture. This includes data collecting through crawling [Raffel et al., 2020] or synthesis [Polu and Sutskever, 2020, Gunasekar et al., 2023], de-duplication [Lee et al., 2021, Kandpal et al., 2022, Tirumala et al., 2023], filtering and selection [Xie et al., 2024a, Albalak et al., 2024], as well as data composition [Xie et al., 2024b] and curriculum [Chen et al., 2024, Ma et al., 2024].

## B.2  Data Selection

Data selection for fine-tuning has been extensively studied, focusing on improving quality [Li et al., 2023c], diversity [Liu et al., 2024], and distribution matching [Li et al., 2023d, Xia et al., 2024, Ni et al., 2024, Kang et al., 2024]. For pretraining, various lightweight filters are utilized [Albalak et al., 2024], including heuristic-based (*e.g.,* language and item count filtering), classifier-based [Brown et al., 2020], and perplexity or loss-based approaches [Wenzek et al., 2019, Qin et al., 2024]. The massive public RedPajama-Data-v2 dataset [Computer, 2023], for example, leverages over 40 quality indicators for data filtering and reweighting. Nevertheless, strict filtering like blocklist [Raffel et al., 2020] and Safety API filtering [Welbl et al., 2021], have been found to hurt evaluation loss or induce bias [Dodge et al., 2021].

Sample-level selection has been extensively studied in previous research [Sener and Savarese, 2017, Killamsetty et al., 2021], particularly through online batch selection [Loshchilov and Hutter, 2015, Schaul et al., 2015, Chang et al., 2017, Katharopoulos and Fleuret, 2018, Jiang et al., 2019]. These approaches have been applied to various classification tasks [Song et al., 2020, Mindermann et al., 2022] and language modeling [Fan and Jaggi, 2023]. However, Kaddour et al. [2023] find that batch selection is not computationally efficient.

Many previous works have employed the general idea of using a reference model as a proxy for data selection. For instance, *Selection Via Proxy* trains a proxy model to select samples with high uncertainty [Coleman et al., 2019]. Xie et al. [2024a] and Engstrom et al. [2024] utilize n-gram models or datamodels with a target dataset to estimate importance weights. Additionally, Xie et al. [2024b] optimize the worst-case excess loss [Oren et al., 2019] relative to a reference model to determine domain weights. One of SLM's scoring functions is excess loss, and the most relevant work related to excess loss is RHO-LOSS [Mindermann et al., 2022], which trains a small model on a holdout set and uses the difference between training loss and holdout loss to select in-batch samples. Although excess loss is mathematically identical to RHO-LOSS, SLM differs in three important ways: 1) The focus is distinct. Motivated by the training dynamics of token loss, the core idea of SLM is to select useful tokens for pre-training. Its score functions are highly flexible and not limited to excess loss (see Appendix H for other functions). In contrast, RHO-LOSS aims to mathematically derive a reducible holdout loss to minimize generalization loss. 2) The meaning and training procedure of the proxy model are different. SLM trains a reference model on high-quality data to reflect the desired data distribution, whereas RHO-LOSS trains a small model on a random holdout set. 3) The selection scale and granularity vary. RHO-LOSS selects sample-level data on a small scale (typically 1K–1M samples) for task-specific fine-tuning tasks such as MNIST [LeCun et al., 1998] and SST-2 [Socher

et al., 2013]. In contrast, SLM conducts fine-grained token-level selection on large-scale language model pre-training, involving up to 80B tokens.

Token-level training strategies have also been explored, especially for the pre-training of BERT-like models using Masked Language Modeling (MLM) [Devlin et al., 2018]. Specifically, "selective masking" involves masking important tokens in the input to focus on learning tokens that are more relevant to downstream tasks [Gu et al., 2020, Lad et al., 2022], whereas "token dropping" aims to reduce training costs by omitting less important tokens [Zhong et al., 2023a, Hou et al., 2022]. [Li et al., 2023e] assesses the quality of each token based on the skewness of its predicted distribution and truncates the noisy tokens during training. Additionally, some research has approached the analysis and detection of under-trained tokens from a tokenization perspective [Rumbelow and Watkins, 2023, Land and Bartolo, 2024]. To our knowledge, we are the first to explore token-level data selection for large language model training, aimed at enhancing data quality and information density at the most fundamental granularity.

### B.3 Language Model Training Dynamics

Investigating the training dynamics of language models is essential for understanding their behavior throughout the training process. This research includes studying internal representations [Saphra and Lopez, 2018], the acquisition of linguistic knowledge [Choshen et al., 2021, Liu et al., 2021], and the phenomenon of grokking [Power et al., 2022]. The analysis by Xia et al. [2022] is the most related to ours, which examines token-level training trajectories in models of varying sizes. Our findings, however, diverge from those of Xia et al. [2022], who posit that tokens with little change in perplexity are "already learned". We identify a spectrum of token patterns, including "easy tokens" and "hard tokens" that resist convergence. Recognizing this, we propose a method of selective language modeling that targets the influential tokens, optimizing the learning process.

### B.4 Scaling Laws

Scaling laws guide us in discovering the impact of factors such as parameter count, data size, and compute on language model performance and behavior. These studies usually focus on predicable scaling though power law [Kaplan et al., 2020, Hernandez et al., 2021], optimal resource allocation [Hoffmann et al., 2022], downstream tasks [Wei et al., 2022b, Isik et al., 2024, Gadre et al., 2024], architectures [Tay et al., 2022], memorization [Tirumala et al., 2022, Carlini et al., 2022, Henighan et al., 2023, Biderman et al., 2024], and repeating data [Hernandez et al., 2022, Muennighoff et al., 2024, Xue et al., 2024]. Most scaling laws on model performance study cross-entory loss on all training tokens, while we focus on the tokens loss of desired distributions.

## C  Limitations and Future Work

**Generalizability**  In math continual pretraining, as depicted in Figure 6, training exclusively with SLM leads to quickly convergence to the domain focused by the reference model, accompanied by a significant rise in the loss of unselected tokens. Although no adverse effects, like biases, have been observed from the increased loss yet, a general pretraining loss on text and code may prevent overfitting [Goodhart and Goodhart, 1984], as suggested by Ouyang et al. [2022] and Azerbayev et al. [2023]. Furthermore, future efforts could broaden the corpus scope of the reference model, and enlarge the pretraining data size, as exemplified by DeepSpeedMath [Shao et al., 2024].

**Scalability**  Due to budget constraints, we have only verified the effectiveness of our method on smaller models (<=7B parameters) and smaller datasets (<100B tokens). Smaller models benefit significantly from removing the loss of irrelevant tokens and focusing on important ones. However, it's possible that very large models trained on extensive corpora may naturally develop this inductive bias to compress useful data (*i.e.,* compressing everything), although it may sounds inefficient for now. Therefore, future works should study whether this selective language modeling technique can scale to very large models and data [Kaplan et al., 2020].

**Is training a reference model necessary?**  To score tokens, we need a high-quality reference model. This could be a base model trained with a small amount of high-quality data, or a performant open-source model. In fact, since we only need input logprobs or perplexity from reference model,

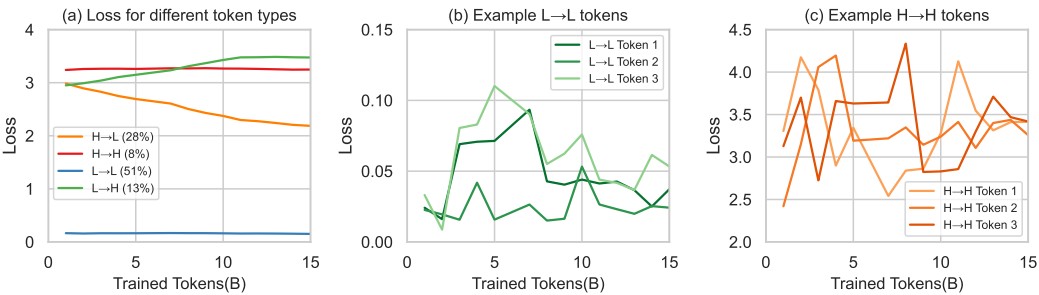

Figure 10: **The loss of four categories of tokens during Mistral-7B pretraining on OpenWebMath.** (a) shows the loss of H→H, L→H, H→L, and L→L tokens during pretraining. (b) and (c) show three cases of fluctuating tokens' loss in L→L and H→H during pretraining, respectively.

we could even utilize more powerful proprietary model APIs. We can input tokens and use the log probabilities of the input returned by the API as reference scores. We leave this for future works.

**How to improve upon SLM?**    There are many natural extensions of SLM, *e.g.,* reweighting tokens instead of selecting may improve robustness; using a reference model as a reward model to guide pretraining with reinforcement learning; adopting multiple reference models to reduce overfitting; designing token-level curriculum learning and iterative strategies for continuous improvements, *etc*.

**Expanding the use of SLM**    SLM may be extended to supervised fine-tuning to address the noise and distribution mismatches in many SFT datasets. Another potential application is alignment, *e.g.,* by training a reference model to emphasize helpfulness, truthfulness, and harmlessness, we may obtain a base model that is natively aligned during the pretraining stage. Meanwhile, we believe that the idea of SLM may find broader applications in multimodal data such as images, videos, and speech, which have a high noise-to-information ratio than text.

# D    Analysis and Visualization of Tokens in Pretraining

## D.1    More Details of Four Categories Tokens

We categorize tokens into four categories: H→H, L→H, H→L, L→L. During the training process, we collected the loss of each token after training on each 1 billion tokens training data. We then used linear fitting and took the difference in loss between the first and last points as evidence of whether the loss decreased during the training process.

Specifically, suppose we have a sequence of token's loss $(l_0, l_1, ..., l_n)$. Our goal is to minimize the sum of the squares of the differences between each data point and its linear predictive value:

$$f(a, b) = \text{minimize} \sum_{i=0}^{n} (l_i - (ax_i + b))^2, \tag{6}$$

where $x_0 = 0$ is the initial checkpoint and $x_n = n$ is the final checkpoint. Substituting these into the fitted equation, we can obtain the Loss values at the start and end after fitting: $\mathcal{L}_{\text{start}} = b$ and $\mathcal{L}_{\text{end}} = an + b$. The change in loss can then be expressed as: $\Delta\mathcal{L} = \mathcal{L}_{\text{end}} - \mathcal{L}_{\text{start}}$. Meanwhile, we represent the average Loss of the last checkpoint as $\mathcal{L}_{\text{mean}}$.

Next, we can classify the tokens based on $\Delta\mathcal{L}$ and the $\mathcal{L}_{\text{mean}}$. We categorize tokens with $\Delta\mathcal{L} < -0.2$ as H→L (loss decreases from high to low) category tokens, and tokens with $\Delta\mathcal{L} > 0.2$ as L→H (loss increases from low to high) category tokens. If $-0.2 \leq \Delta\mathcal{L} \leq 0.2$ and $l_n \leq \mathcal{L}_{\text{mean}}$, then tokens are classified as L→L (loss remains low); if $l_n > \mathcal{L}_{\text{mean}}$, they are classified as H→H (loss remains high). In Figure 10, we have added the tokens' loss curves of the 7B model which is consistent with the other experimental settings in §2.1, for readers to refer to whether similar phenomena exist on larger models. In Figure 11, we visualize examples of the four categories of tokens in actual text.

### D.2 Non-Converging Tokens in Pretrainig

In §2.1, we mentioned that during the training process, only a minority of tokens belong to the H→L category. Among the remaining categories of H→H and L→L tokens, there are tokens that exhibit significant fluctuations during training. Furthermore, there are instances where H→L tokens are not effectively learned. Therefore, in our analysis, we specifically select those tokens from these categories that demonstrate considerable variability and distinct loss. We visualize these tokens that exhibit abnormal behavior during the training process. As illustrated in Figure 12, we find that the majority of these tokens originate from rather chaotic corpora. For instance, the corpora may include a mix of custom symbols, unintelligible gibberish, and information such as timetables and bibliographic references. Within a segment of normal text, there may also be fluctuations in the usage of common conjunctions, word suffixes, and punctuation marks. The latter may not necessarily be disastrous for training; in fact, it could represent a normal occurrence. However, if we can effectively mitigate the losses caused by the former, it might lead to more stable and efficient model training.

## E Evalution Details

### E.1 Math Evalution

We conducted a comprehensive evaluation of the model across various math reasoning benchmarks, encompassing a range of difficulties from elementary to university level, multiple mathematical domains, and diverse question types including multiple-choice and open-ended questions. Our benchmarks include GSM8k [Cobbe et al., 2021], MATH [Hendrycks et al., 2021], GSM-Hard [Gao et al., 2022], SVAMP [Patel et al., 2021], ASDIV [Miao et al., 2020], MAWPS [Koncel-Kedziorski et al., 2016], TabMWP (TAB) [Lu et al., 2023], MathQA (MQA) [Amini et al., 2019], MMLU-STEM [Hendrycks et al., 2020], and SAT [Azerbayev et al., 2023].

### E.2 General Evalution

In the evaluation of general domain, we followed the lm-evaluation-harness [Gao et al., 2023] and evalute model on MMLU [Hendrycks et al., 2020], BBH [Suzgun et al., 2022], AGIEval [Zhong et al., 2023b], ARC-Easy and ARC-Challenge [Clark et al., 2018], BoolQ [Clark et al., 2019], PIQA [Bisk et al., 2020], Hellaswag [Zellers et al., 2019], WinoGrande [Sakaguchi et al., 2021], OpenBookQA [Mihaylov et al., 2018]. On HumanEval [Zheng et al., 2023] and TydiQA [Clark et al., 2020], we follow the evaluation pipeline of open-instrcut [Ivison et al., 2023] and report Pass@1 and Pass@10 for HumanEval and F1 for TydiQA. For MBPP [Austin et al., 2021] benchmark, we follow the evaluation pipeline of DeepSeek-Coder [Guo et al., 2024], and report Pass@1 and Pass@10.

## F Relate the Selected Tokens' Loss to Downstream Task Performance

In this section, we declare the details about correlating the loss of selected tokens with the performance of downstream tasks. Concurrent study has explored similar methods to study the impact of scaling laws with the performance of models in downstream tasks [Gadre et al., 2024]. Our analysis here differs in that it aims to elucidate the relationship between the decrease/increase in loss for selected/unselected tokens and the model's performance on downstream tasks.

We use the average accuracy of MATH and GSM8K as the standard for measuring downstream tasks performance of model. Based on the trend of data points in Figure 7, we propose the relationship between the average accuracy of downstream tasks and selected/unselected tokens' loss,

$$Acc(\mathcal{L}) = \log(a * \mathcal{L} + c) \tag{7}$$

The parameters $a$ and $c$ are fitted from the data. If the loss of selected tokens $\mathcal{L}_s$ is used for fitting, then $a > 0$. Conversely, if the loss of unselected tokens $\mathcal{L}_{us}$ is used for fitting, then $a < 0$. Therefore, we believe that training the model on selected tokens can effectively improve its performance on downstream tasks, while unselected tokens may have a detrimental effect on the model's performance in downstream tasks.

Table 4: **Full Self-Reference results on Tinyllama-1.1B.**

| Score Function | Select Ratio | GSM8K | MATH | SVAMP | ASDiv | MAWPS | MQA | AVG |
|---|---|---|---|---|---|---|---|---|
| - | 100% | 6.3 | 2.6 | 21.7 | 36.7 | 47.7 | 13.9 | 21.5 |
| $\mathcal{L}_{\text{RM}}(x_i)$ | 90% | **7.4** | 4.4 | **23.4** | 38.7 | 51.9 | **14.4** | 23.4 |
| | 80% | 6.4 | **4.6** | 23.1 | 39.7 | 52.0 | 14.3 | 23.4 |
| | 70% | 6.7 | **4.6** | 23.3 | **40.0** | **54.5** | 14.3 | **23.9** |
| | 60% | 7.0 | **4.6** | 22.2 | 38.5 | 52.2 | 13.7 | 23.0 |
| | 50% | 5.7 | 4.2 | 20.7 | 36.7 | 46.7 | 10.3 | 20.7 |
| $\mathcal{H}_{\text{RM}}(x_i)$ | 90% | 6.7 | 3.0 | 23.7 | 40.3 | 52.3 | 13.1 | 23.2 |
| | 80% | 6.8 | 3.6 | 22.5 | **40.6** | **52.9** | 13.6 | 23.3 |
| | 70% | **7.0** | 4.8 | 23.0 | 39.3 | 50.5 | 13.5 | 23.0 |
| | 60% | 6.5 | 4.8 | **26.5** | 37.3 | 49.7 | **15.6** | **23.4** |
| | 50% | 4.7 | **5.8** | 20.9 | 33.8 | 42.5 | 11.1 | 19.8 |
| $\mathcal{H}_{\text{RM}}(x_i) \cup \mathcal{L}_{\text{RM}}(x_i)$ | $50\% \cup 70\%(80\%)$ | 6.4 | 3.6 | 22.7 | 38.4 | 52.6 | 15.3 | 23.2 |
| | $70\% \cup 60\%(77\%)$ | 6.3 | 4.6 | 24.4 | 39.6 | 51.4 | **16.3** | **23.8** |
| | $70\% \cup 50\%(75\%)$ | 6.9 | 5.6 | 23.2 | **39.9** | **52.9** | 12.6 | 23.5 |
| | $60\% \cup 60\%(70\%)$ | 6.7 | 5.2 | **24.7** | 39.2 | 50.6 | 14.6 | 23.5 |
| | $60\% \cup 50\%(68\%)$ | 7.1 | 5.8 | 21.7 | 37.3 | 49.6 | 15.3 | 22.8 |
| | $60\% \cup 40\%(65\%)$ | **7.3** | **6.0** | 23.6 | 36.9 | 48.6 | 13.1 | 22.6 |
| $\mathcal{H}_{\text{RM}}(x_i) \cap \mathcal{L}_{\text{RM}}(x_i)$ | $80\% \cap 90\%(76\%)$ | 6.0 | 4.4 | 23.7 | 38.5 | 51.2 | 13.3 | 22.8 |
| | $75\% \cap 75\%(72\%)$ | 7.8 | 5.2 | **24.2** | 39.4 | **54.9** | 14.7 | 24.4 |
| | $70\% \cap 90\%(68\%)$ | 6.8 | 4.6 | 22.2 | 40.3 | 53.0 | 14.8 | 23.6 |
| | $80\% \cap 80\%(67\%)$ | **8.2** | **6.4** | 21.2 | 39.1 | 53.4 | 15.0 | 23.9 |
| | $70\% \cap 70\%(60\%)$ | 7.1 | 5.0 | 23.5 | **41.2** | 53.8 | **18.0** | **24.8** |

## G Examples of Tokens Selected by SLM

### G.1 Token Selected Examples

In Figure 13, we present several examples of tokens selected by the SLM method, with content marked in blue indicating the tokens actually chosen during the pretraining process.

### G.2 Dynamic Token Selected

In Figure 14, we display the dynamic changes in token selection tendencies throughout the SLM training process. We chose four checkpoints during the training process (0%, 33%, 66%, and 100%) to analyze the current tendencies in token selection. The preferences for token selection are indicated by different colors, ranging from high to low preference, typically represented as deep blue, blue, black, orange, and dark orange, respectively.

## H Self-Reference Setting

In this section, we will provide a detailed introduction to the reference loss score function and information entropy score function in SLM. Reference loss score function is to directly use the loss of the reference model as the basis for selecting tokens. The higher the token's loss of the reference model, the lower the expectation that the token will be selected. The score $\mathcal{L}_{\text{RM}}(x_i)$ can be directly obtained by referring to Equation 1. Information entropy score function is to select the corresponding token based on the information entropy of the reference model in each token. The information entropy of token $x_i$ can be expressed as:

$$\mathcal{H}_{\text{RM}}(x_i) = -\sum_{k=1}^{V} P(t_k|x_{<i}) \log P(t_k|x_{<i}), \qquad (8)$$

Table 5: **Weak-to-Strong generalization result on math benchmark.**

| Model | Train Toks | GSM8K | MATH | SVAMP | ASDiv | MAWPS | TAB | MQA | MMLU STEM | SAT | AVG |
|---|---|---|---|---|---|---|---|---|---|---|---|
| Llama-2-7B-CT | 15B | 28.4 | 13.6 | 50.3 | 62.8 | 79.5 | 37.6 | 34.1 | 41.6 | 43.5 | 43.5 |
| Llama-2-7B-CT w/ 1B RM | 10.5B | 29.8 | 16.0 | 55.5 | 63.7 | 80.4 | 37.9 | 34.3 | 38.2 | 43.8 | 44.4 |

where $t_k$ represents the i-th token in the vocabulary, and $V$ represents the size of the vocabulary. The intuition of this strategy is that the higher the information entropy, the higher the uncertainty of the token in the context. Therefore, we consider that if the language model is still uncertain for certain tokens after pretraining, we do not expect that the language model will learn it during pretraining. In Table 4, we provide more SLM results, including different select ratios and combinations of two score functions, for the convenience of the readers to refer to.

## I  Weak-to-Strong Generalization

Apart from the main experiments where we use the same base model for the reference and continual pretraining, we also investigate if a smaller reference model can effectively guide the pretraining of a larger model. We use Tinyllama-1.1B as reference model and continual pretraining Llama-2-7B on 15B OpenWebMath tokens. Results presented in Table 5 indicate that, despite the considerable gap between the small and large models [Li et al., 2023f], employing the small reference model to token selection can still yield benefits to the pre-training of the larger model. If reference and training models have different vocabularies, one can consider performing token alignment [Wan et al., 2024, Fu et al., 2023], which we leave for future work.


## Examples of Four Categories of Tokens

GMAT 1: 670 Q49 V31 \n GMAT 2: 710 Q50 V35 \n Followers: 175 \n \n Kudos [?]: 890 [0], given: 235 \n \n Re: Mr. and Mrs Wiley, VIC[#permalink] 13 Feb 2010, 01:03 \n Ans A \n \n their first child was born after J years... \n \n thus 1 child —> j years \n \n => thus after another J years his age = J \n \n thus his age is J –> after 2J years and 2j after 3j years \n \n his present age is T which is after T years. \n \n thus total time after 2years will be T+2 \n since after every J year they have a child after T+2 they will have \frac{(T+2)}{J} + 1 ( +1 is for the oldest) \n \n thus A \n _________________ \n \n Fight for your dreams :For all those who fear from Verbal- lets give it a fight \n \n Money Saved is the Money Earned \n \n Jo Bole So Nihaal , Sat Shri Akaal \n \n Gmat test review : \n 670-to-710-a-long-journey-without-destination-still-happy-141642.html \n \n Intern \n Joined: 06 Apr 2012 \n Posts: 28 \n Followers: 0 \n Kudos [?]: 4 [0], given: 37 \n \n Re: Mr. and Mrs Wiley, VIC[#permalink] 21 Nov 2012, 07:46 \n jeeteshsingh wrote: \n Need the solution using Algebra.... \n \n Mr. & Mrs Wiley have a child every J years. Their oldest child is now T years old. If they have a child 2 years from now, how many children will they have in total? \n \n (A) \frac{T+2}{J} + 1 \n \n (B) JT + 1 \n \n (C) \frac{J}{T} + \frac{1}{T} \n \n (D) TJ - 1 \n \n (E) \frac{T+J}{J} \n \n [Reveal] Spoiler: OA: \n (A) \n \n Source: Manhattan Guide \n \n Bunuel - would really appreciate you providing your bit on solving the original problem above algebraically. The problem and various explanations remain confusing. Should we think of it as a progression or some other way? Please share your take. Thank you. \n Veritas Prep GMAT Instructor \n Joined: 16 Oct 2010 \n Posts: 4566 \n Location: Pune, India \n Followers: 1029 \n \n Kudos [?]: 4460 [1] , given: 162 \n \n Re: Mr. and Mrs Wiley, VIC[#permalink] 21 Nov 2012, 09:45 \n 1 \n KUDOS \n Expert's post \n jeeteshsingh wrote: \n Need the solution using Algebra.... \n \n Mr. & Mrs Wiley have a child every J years. Their oldest child is now T years old. If they have a child 2 years from now, how many children will they have in total? \n \n (A) \frac{T+2}{J} + 1 \n \n (B) JT + 1 \n \n (C) \frac{J}{T} + \frac{1}{T} \n \n (D) TJ - 1 \n \n (E) \frac{T+J}{J} \n \n [Reveal] Spoiler: OA: \n (A) \n \n Source: Manhattan Guide \n \n Think of it as an Arithmetic Progression where every subsequent term (child) has a difference of J yrs from the previous term (child). \n \n 1st child, 2nd child, 3rd child, ....... nth child (to be born after 2 yrs) \n \n What is the difference between first and last terms (children)? (T + 2) yrs \n \n What is the common difference (age difference between two consecutive kids)? J yrs \n \n What is the number of terms (children)? (T + 2)/J + 1 \n (Number of terms of an AP is n = (Last term - First term)/Common Difference + 1. ) \n _________________ \n \n \n Karishma \n Veritas Prep | GMAT Instructor \n My Blog \n \n Save $100 on Veritas Prep GMAT Courses And Admissions Consulting Enroll now. Pay later. Take advantage of Veritas Prep's flexible payment plan options. Veritas Prep Reviews Re: Mr. and Mrs Wiley, VIC [#permalink] 21 Nov 2012, 09:45 Similar topics Replies Last post Similar Topics: 1 Mr. and Mrs. O'Leary (SC) 5 08 Jul 2012, 07:15 Mr. INVESTOR invested a total of $12,000 for a one-year 4 30 Mar 2007, 09:24 \n 2 Mr. and Mrs. Wiley have a child every J years. Their oldest 7 19 Feb 2007, 11:40 \n Mr.kevincan 6 16 Aug 2006, 12:26 \n PS: Mr. & Mrs. Smith 2 06 Dec 2005, 00:03 \n Display posts from previous: Sort by Sciencemadness Discussion Board » Fundamentals » Reagents and Apparatus Acquisition » Sulphuric Acid in Australia Select A Forum Fundamentals  » Chemistry in General  » Organic Chemistry  » Reagents and Apparatus Acquisition  » Beginnings  » Responsible Practices  » Miscellaneous  » The Wiki Special topics  » Technochemistry  » Energetic Materials  » Biochemistry  » Radiochemistry  » Computational Models and Techniques  » Prepublication Non-chemistry  » Forum Matters  » Legal and Societal Issues \n \n Pages: 1 2 \n Author: Subject: Sulphuric Acid in Australia \n hissingnoise \n International Hazard \n \n Posts: 3939 \n Registered: 26-12-2002 \n Member Is Offline \n \n Mood: Pulverulescent! \n \n I've stated several times on various threads, that $SO_3$ produces a practically incondensable acid mist when led to water and, BTW, at 700°C the decomposition rate of $SO_3$ is ˜87% . . . \n Cracking $Na_2S_2O_7$ proceeds at ˜466°C and the issuing gasses are readily absorbed by conc. $H_2SO_4$ to form oleum! \n \n Phthalic Acid \n Harmless \n \n Posts: 19 \n Registered: 7-8-2011 \n Location: Australia \n Member Is Offline \n \n Mood: No Mood \n \n That's a good idea Neil, I'll be sure to try that next time (probably for H2O2). Just went to Tradelink and asked if they sold Moflo drain cleaner. The guy said yeah and I asked for a liter of it. No problems whatsoever, he just said "be careful with it". It was $45 but a liter will last me a while and making it myself would've been vastly more expensive I imagine. Success! MeSynth Hazard to Others Posts: 107 Registered: 29-7-2011 Member Is Offline Mood: http://www.youtube.com/watch?v=5ZltqIVuDIo Sulfuric acid can be produced in the laboratory by burning sulfur in air and dissolving the gas produced in a hydrogen peroxide solution. SO2 + H2O2 → H2SO4 this was found on wikipedia... did you not look through the sullfuric acid wiki before boiling down batery acid? anyways... There are some good videos on youtube that demonstrate how to synthesize sulfuric acid using different methods. The drain cleaner you get from the store will be impure and may contain organic matter that discolors the acid.

Figure 11: **Sample text containing four categories of tokens.** Among them, blue represents tokens of categorie H→L, green indicates tokens of categorie L→L, yellow signifies tokens of categorie H→H, and red denotes tokens of categorie L→H.

**_Examples of Tokens that Exhibit Abnormal Behavior during Training_**

as \n  \n \begin{aligned} A \in \{\pm \begin{bmatrix}\cos\theta & - \sin\theta \\ \sin\theta & \cos\theta \\ \end{bmatrix}, \pm \begin{bmatrix}\cos\theta & \sin\theta \\ \sin\theta & - \cos\theta \\ \end{bmatrix}, \pm \begin{bmatrix}i \sinh\theta & -\cosh\theta \\ \cosh\theta & i \sinh\theta \\ \end{bmatrix}, \pm \begin{bmatrix}i \sinh\theta & \cosh\theta \\ \cosh\theta & - i \sinh\theta \\ \end{bmatrix}\}\end{aligned} \quad\quad\quad(25) \n  \n I suspect this class of transformations has a name in the grand group classification scheme, but I don't know what it is. ### Mathematics Class XI \n  \n Unit-I: Sets and Functions \n Chapter 1: Sets \n Unit-II: Algebra \n Chapter 5: Binomial Theorem \n Chapter 6: Sequence and Series \n Unit-III: Coordinate Geometry \n Chapter 1: Straight Lines \n Chapter 2: Conic Sections \n Unit-IV: Calculus \n Unit-V: Mathematical Reasoning \n Unit-VI: Statistics and Probability \n Chapter 1: Statistics \n Chapter 2: Probability \n # Graphs of Trigonometric Functions \n  \n (i)

Geometry in any field. Queries are case-independent. Funct* Wildcard queries are specified by * (e.g. functions, functorial, etc.). Otherwise the search is exact. "Topological group" Phrases (multi-words) should be set in "straight quotation marks". au: Bourbaki & ti: Algebra Search for author and title. The and-operator & is default and can be omitted. Cheb yshev | Tschebyscheff The or-operator | allows to search for Cheb yshev or Tschebyscheff. "Quasi* map*" py: 1989 The resulting documents have publication year 1989. so: Eur* J* Mat* Soc* cc: 14 Search for publications in a particular source with a Mathematics Subject Classification code (cc) in 14. "Partial diff* eq*" ! elliptic The not-operator ! eliminates all results containing the word elliptic. dt: b & au: Hilbert The document type is set to books; alternatively: j for journal articles, a for book articles. py: 2000-2015 cc: (94A | 11T) Number ranges are accepted. Terms can be grouped within (parentheses). la: chinese Find documents in a given language. ISO 639-1 language codes can also be used.

Code: Select all \n  \n x = 64, y = 86, rule = B3/S23 \n 13bo$3bobo6bo$4b2o6b3o$4bo$54bo$54bobo$13b2o39b 2o  $12b2o44b2o$3o11bo43b  \n  o3b2o$2bo49bo  6bo2bo$bo50b  2o6b  obo$51bob  o7bo$7bo49bo$7 b3o47b3o$10bo5b2o     \n     42bo$9b2o4b2o     42b     2o$17bo7$13bo$3b     obo6bo$4b     2o6b 3o$4bo$54bo$54b     obo$13b     2o    39b2o$12b2o44b2o$3o11bo43bo3b2o$2bo49bo6bo2bo$bo50b    2o6b obo$51bobo7bo$7bo49bo$7b3o47b3o$10bo5b2o42bo$9bo5b2o42bo$9b2o6bo41b2o7$13bo$3bobo6bo$4b 2o6b3o$4bo$54bo$54bobo$13b2o39b2o$12b2o44b2o$3o11bo43bo3b2o$2bo49bo 6bo2bo$bo50b2o6bobo$51bobo7bo$7bo49bo$7b3o47b3o$10bo5b2o42bo$7b3o5b2o

40b3o$7bo9bo39bo7$13bo$3b  obo6bo$4b  2o6b  3o$4bo$54bo$54b  obo$13b  2o39b  2o$  \n  12b  2o44b 2o$3o11bo43bo3b  2o$2bo49bo6bo2bo$bo50b  2o6b obo$51b  obo7bo$7bo49bo$7b  3o47b  3o$10bo5b  2o42bo$7b 2obo4b  2o40b 2obo$7b obo7bo39b obo! The 16-bitter thus goes down to 9 gliders. It does not reduce any further 17-bitters, though. Princess of Science, Parcly Taxel Kazyan Posts: 867 Joined: February 6th, 2014, 11:02 pm ### Re: 17 in 17: Efficient 17-bit synthesis project Good catch, Sokwe. #61 in 15G:

Ground Penetrating Radar for Archaeology \n  \n Workshop | December1 | 1-5 p.m.  | 1012251 College (Archaeological Research Facility) \n  \n Scott Byram, Research Associate, Archaeological Research Facility, UC Berkeley \n  \n Archaeological Research Facility \n  \n At 1pm the workshop will begin at the UC Faculty Club lawn where subsurface features are being mapped. \n  \n ### Student Probability/PDE Seminar: Large Deviation Principle for random graphs II \n  \n Seminar | December1 | 2:10-3:30 p.m. | 891Evans Hall \n  \n Fraydoun Rezakhanlou, UC Berkeley \n  \n Department of Mathematics \n  \n ### BLC Fellows Forum \n  \n Presentation | December1 | 3-5 p.m. | Dwinelle Hall, B-4 (Classroom side) \n  \n FAll 2017 BLC Fellows, UC Berkeley \n  \n Berkeley Language Center \n  \n Teaching French Listening Comprehension and Cultural Awareness through Regional Variation \n Elyse Ritchey, GSR, French \n At the university level, French language instruction in the US traditionally includes a course on phonetics and pronunciation. While the major aim of such courses is to improve students' speaking and listening competence, they also emphasize speaking 'correctly' using... More > \n  \n ### MENA Salon \n \n Workshop | December1 | 3-4 p.m. | 340Stephens Hall \n  \n Every Friday in the semester, the CMES hosts an informal week

Figure 12: **An example of an abnormal state of token perplexity during pretrainig process.** The tokens highlighted in orange represent tokens that were significant abnormalities during the pretraining process.

***Token Selected Examples***

• Process the student answer as a Math Object Formula, and break down its parse tree by its top-level operators. The idea is to create an array of the student's primitive factors, so say 3(x+1)(x+2)^2 gives (3,x+1,x+2). • Because we may want factoring over Z, checking the gcd of coefficients within each factor. • Pass each of these things to SAGE and ask if the nonconstant factors are reducible over Z or Q. Also ask if they are monic. These things at least we learned how to do at the Vancouver code camp. The end goal is to count the following forms as correct, possibly controlled by flags: n \{}prod (factor)^power, where each factor is irreducible in Z[X], n in Z r \{}prod (factor)^power, where each factor is irreducible and monic in Q[X], r in Q I suppose on the last one the monic requirement could be dropped with a flag. I have no plans to check that the form is fully condensed, e.g. forcing (x+1)^2 and rejecting (x+1)(1+x)

The equation of the path traversed by a projectile is called equation of trajectory. \n \n Suppose, the body reaches the point P after time ( t ) . \n \n Horizontal motion has no acceleration. Thus, using kinematic equation, horizontal distance covered will be – \n \n x = u \cos \theta t \n \n Or, \quad t = ( \frac { x }{ u \cos \theta } ) \n \n Vertical motion has constant acceleration ( g ) . Thus, distance covered will be – \n \n y = ( u \sin \theta ) t - \left ( \frac {1}{2} \right ) g t^2 \n \n = ( u \sin \theta ) \left ( \frac {x}{u \cos \theta} \right ) - \left ( \frac {1}{2} \right ) g \left ( \frac {x}{u \cos \theta} \right )^2 \n \n = \left ( \tan \theta \right ) x - \left ( \frac {g}{2 u^2 \cos^2 \theta} \right ) x^2 \n \n In this equation, ( \theta, \ u \ \text {and} \ g ) are constants. Thus, \n \n 1. Term \left ( \tan \theta \right ) is a constant, let it is ( p ) \n 2. Term \left [ \left ( \frac {g}{2 u^2 \cos^2 \theta} \right ) \right ] is also a constant, let it is ( q ) \n \n So, \quad y = p x - q x^2 \n \n Therefore, ( y \propto x^2 ) , which is a required condition of a parabola. The trajectory of the projectile is a parabola. \n \n ### Time of Maximum height \n \n As the body is projected it goes up. Vertical component of velocity ( u \sin \theta ) gradually diminishes and becomes zero at the maximum height of flight. After that, body starts moving downwards. \n \n Let, ( t_m ) is the time to reach at maximum height ( h_m ) of flight. \n \n Therefore, from kinematic equation, we have – \n \n 0 = u \sin \theta - g t_m \n \n Or, \quad t_m = \left ( \frac {u \sin \theta}{g} \right ) \n \n ### Time of Flight \n \n Total time taken by the projectile between the instant it is projected and till it reaches at a point in the horizontal plane of its projection is called Time of flight. \n \n Let, the body reaches at point B on ground after time ( T_f ) of projection. Then – \n \n Net vertical displacement covered during the time of flight is zero. Using kinematic equation of motion, we get – \n \n 0 = ( u \sin \theta ) T_f - \left ( \frac {1}{2} \right ) g \ ( T_f )^2 \n \n Or, \quad T_f = \left ( \frac {2 u \sin \theta}{g} \right ) = 2 \left ( \frac {u \sin \theta}{g} \right ) \n \n = 2 t_m \n \n Thus, \quad \text {Total time of flight} = \text {Time of ascent} + \text {Time of descent} \n \n = 2 \times \text {Time of maximum height.} \n \n ### Maximum height of Flight \n \n It is the maximum height reached by a projectile. It is denoted by ( h_m ) \n \n At the highest point of flight, the vertical component of velocity becomes zero. \n \n From kinematic equation of motion, we have – \n \n v^2 = u^2 + 2 a s \n \n Therefore, \quad 0^2 - ( u \sin \theta )^2 = 2 ( - g ) h_m \n \n Or, \quad h_m = \left ( \frac {u^2 \sin^2 \theta}{2 g} \right )

We identify two equations having the same solution with the equivalence relation: \n \n $(a,b) \sim (c,d) \mbox{ if and only if } ad = bc$ \n \n To show that this is an equivalence relation: \n \n 1. Reflexivity: $$(a,b) \sim (a,b)$$ if and only if $$ab = ba$$ which is true. Hence it is reflexive. \n 2. Symmetry: $$(a,b) \sim (c,d)$$ if and only if $$ad = bc$$ if and only if $$bc = ad$$ if and only if $$(c,d) \sim (a,b)$$. Hence it is symmetric. \n 3. Transitivity: $$(a,b) \sim (c,d)$$ and $$(c,d) \sim (e,f)$$ if and only if $$ad = bc$$ and $$cf = de$$. Multiplying these equations together, we get $$adcf = bcde$$. We can cancel $$d$$ and $$c$$ from both sides to get $$af = be$$. Hence $$(a,b) \sim (e,f)$$. \n \n Hence, we have successfully formed the set of rational numbers when we factor out the equivalence classes! \n \n $\mathbb{Q} = \frac{\mathbb{Z} \times \mathbb{Z}\backslash\{0\}}{\sim}$ \n \n Let's now take a look at what members of $$\mathbb{Q}$$ look like, say for the equation $$2x = 3$$. This equation is represented by the ordered pair

If the light moves in a purely radial direction, we can describe its path by the coordinate functions $$t(\lambda)$$ and $$r(\lambda)$$. The equation of motion $$ds^2 =0$$ then takes the form $$g_{tt} \left(\frac{dt}{d\lambda}\right)^2 + g_{rr} \left(\frac{dr}{d\lambda}\right)^2 = 0,$$ which we can rewrite as $$\left(\frac{dt}{dr}\right)^2 = -\frac{g_{rr}}{g_{tt}}.$$ \n \n The length of the rod is then $$L = c \int_{r_1}^{r_2} \frac{dt}{dr} \text{ d}r = c \int_{r_1}^{r_2} \sqrt{-\frac{g_{rr}}{g_{tt}}} \text{ d}r,$$ where I have taken the positive square root because $$r_2 > r_1$$. \n \n Notice that the length is independent of the signature of the metric, so whether you work with the (-+++) or (+—) metric is purely conventional and will not change the physics. \n \n For the Schwarzschild metric, we obtain explicitly $$L = r_2 - r_1 + r_s \ln\left(\frac{r_2 - r_s}{r_1 - r_s}\right) > r_2 - r_1.$$ \n \n Now what happens if you magically, instantaneously increase the mass of the black hole? I think the length $$L$$ of the rod stays the same (I'm here assuming that the rod is infinitely stiff), but that it would now "appear shorter" to the distant observer - i.e. it would no longer occupy the entire space between $$r_1$$ and $$r_2$$.

Figure 13: **Specific examples of selecting tokens during the selective pretraining process of the RHO-1.** The tokens marked in blue represent the actual tokens trained during the training process, while the remaining black tokens are not trained during the training process.

```
After Training 0% Checkpoint
```

Item Type: Journal Article Copyright of this article belongs to Elsevier. Division of Mechanical Sciences > Mechanical Engineering 28 May 2007 19 Sep 2010 04:36 http://eprints.iisc.ernet.in/id/eprint/10277 # Question #8de97 \n \n Dec 10, 2016 \n \n That is not an identity. \n \n #### Explanation: \n \n Recall that \n \n ${\cot}^{2} x + 1 = {\csc}^{2} x$. \n \n So, we can write \n \n $\frac{1 - {\csc}^{2} x}{\csc} ^ 2 x = \frac{1 - \left({\cot}^{2} x + 1\right)}{\csc} ^ 2 x$ \n \n $= {\cot}^{2} \frac{x}{\csc} ^ 2 x$ \n \n Recall also that $\cot x = \cos \frac{x}{\sin} x$ and $\csc x = \frac{1}{\sin} x$. \n \n This allows us to continue \n \n $= \frac{{\cos}^{2} \frac{x}{\sin} ^ 2 x}{\frac{1}{\sin} ^ 2 x}$ \n \n $= {\cos}^{2} \frac{x}{\sin} ^ 2 x \cdot {\sin}^{2} \frac{x}{1}$ \n \n $= {\cos}^{2} x$ \n \n Which is not identically $\cos x$. \n \n (${\cos}^{2} x = \cos x$ only when $\cos x = 1$ or $0$) \n \n Dec 10, 2016 \n \n No. It is equal to ${\sin}^{2} x - 1$. \n \n #### Explanation: \n \n If we have $\frac{1 - x}{x}$, we can write it as $\frac{1}{x} - \frac{x}{x}$. \n \n The same way, $\frac{1 - {\csc}^{2} x}{\csc} ^ 2$ can be written as $\frac{1}{\csc} ^ 2 x - \frac{{\csc}^{2} x}{{\csc}^{2} x}$. \n \n This is equal to ${\sin}^{2} x - 1$. SMS scnews item created by Hannah Bryant at Wed 25 May 2022 1227 \n Type: Seminar \n Distribution: World \n Expiry: 31 May 2022 \n Calendar1: 31 May 2022 1500-1600 \n CalLoc1: Quad S224 & via Zoom \n CalTitle1: SMRI 'What is ...a virtual knot?' Hans Boden (McMaster University) \n Auth: hannahb@w1d4n6z2.staff.sydney.edu.au (hbry8683)

Figure 14: **An example of dynamic token selection changes during the training process**, which illustrated with five different score levels represented by deep blue, light blue, black, light orange, and dark orange. The bluer the color indicates a higher tendency for the token to be selected, while the more orange the color suggests a lower tendency for the token to be selected.

