# OpenReview forum: "Not All Tokens Are What You Need for Pretraining"
_NeurIPS.cc/2024/Conference — NeurIPS 2024 oral_

### Official Review · Reviewer_pLtQ · 2024-07-05

**Soundness:** 3
**Presentation:** 3
**Contribution:** 4
**Rating:** 7
**Confidence:** 4

**Summary:**

The paper analyzes token-level training dynamics in continued pretraining, identifying four loss patterns: persistent low loss, persistent high loss, increasing loss, and decreasing loss. Motivated by these patterns, the paper proposes a modification to language modeling called Selective Language Modeling (SLM), which only trains on a subset of the input tokens. This subset is selected by training a high-quality reference model and computing the "excess loss" of the target model -- i.e., the token-level difference between the target model and reference model loss. The model trained using SLM, Rho, achieves strong performance on math and other benchmarks relative to a model using normal continual pretraining.

**Strengths:**

S1. The four categories of token-level loss are interesting and (to the best of my knowledge) not a previously noted phenomenon. The authors provide an interesting analysis of this phenomenon, and use it to motivate their method.

S2. The idea of selecting a subset of tokens to train on is clever and appears effective. The results, particularly on math benchmarks, after continual pretraining with SLM are quite strong and compared to sensible baselines.

S3. The analysis is relatively comprehensive and contains several interesting points, especially the section comparing the correlation between token losses and downstream performance for selected/unselected tokens.

**Weaknesses:**

W1.  *Concerns about training time/cost*. The "10x faster"/"5x faster" claims in Figure 1 don't factor in the cost of pre-scoring each token by both the reference and training model. Can you measure and report this cost? It seems like what the figure actually shows horizontally is *data efficiency*, not speed of training. More generally, I think the claims about efficiency need to be more clearly explained-- the *data* efficiency claim is well-supported, but "efficient" used more broadly (e.g. lines 83, 114-115, 205) generally suggests a time or space efficiency claim, which I don't think the paper supports (or even really intends to claim).

W2. *Scope of claims in title/abstract*. The title and start of the abstract suggest that the method is meant to be applied throughout pretraining, but the paper focuses on continued pretraining. Additionally, the eval focuses predominately on math datasets, which involve many tokens which may be relatively infrequent in pretraining corpora but frequent in-domain. This seems like the ideal domain for this kind of strategy--and, as Figure 5 shows, the gains are much more modest for other tasks. It seems the main finding is that "SLM is a strong method for continual pretraining for math tasks (and slightly beneficial for general domain tasks)", but the title/first 10 lines seem to suggest "SLM should be used instead of CLM for pretraining from scratch," which isn't supported or claimed elsewhere in the paper.

W3. The reference model should be included in the results tables as well. Does Rho outperform the reference model used for token selection?

W4. Doing hard selection cutoffs seems a bit heavy-handed; it's possible that weighting examples according to their "excess loss" might lead to higher performance. The authors do mention this as a future direction in the appendix.

**Questions:**

Q1a. The contents of each token category seem quite important to the paper’s stated motivation of removing noisy tokens from pretraining. Can you provide example sets of tokens / themes?

Q1b. It's not really clear from the few examples provided in Figure 11-14 how to interpret these four loss categories. Are there specific tokens which are generally in one category regardless of the document they occur in (e.g. very rare tokens, or numbers in math equations)?

Q1c. Figure 14: What conclusion should we draw from the differences in tokens selected over time? It's hard to interpret this figure.

Q2. What artifacts do you plan to release? In particular, do you plan to release the model Rho? The 0.5 B and 1.9B datasets you compiled? Checkpoints trained on increasing selection percentages?

Q3. I understand if this is not possible to address in the rebuttal period, but I'm curious if using this method has any impact on the downstream memorization of pretraining data.

Other suggestions/line comments (no need to address in rebuttal):
* Line 33: "limiting LLM's potential to merely mediocre intelligence" is a pretty meaningless phrase -- what does "mediocre intelligence" mean? I suggest revising to be more specific about the claim here (e.g. "limiting the model's capabilities").
* Figure 8 is hard to understand
* Figure 11 is not colorblind-friendly
* Line 764: typo in spelling of Tinyllama

**Limitations:**

Limitations listed look reasonable.

---

> ### Author Rebuttal · Authors · 2024-08-07
>
> Dear Reviewer pLtQ,
>
> Thank you for your detailed review and thoughtful feedback, especially for finding our token-level loss research novel, our SLM idea clever and effective, and our analysis comprehensive!
>
> ### **W1. Data efficiency vs. Training time**
>
> Thank you for your suggestion! We believe that the term "data efficiency" better captures the essence of the SLM method we aim to present. Therefore, in the revised version, we will highlight the advantages of the method from the perspective of data efficiency.
>
> Additionally, it is important to clarify that we only need to perform a single forward pass for scoring, which is significantly faster than the training process. During the training stage, the loss score is automatically calculated in the forward pass of the training model, so there are no additional costs during training.
>
> ### **W2. Scope of claims**
>
> Thank you for your feedback. Our method is initially designed for pre-training. However, due to budget constraints, we were unable to conduct large-scale pre-training from scratch. We will leave the exploration of SLM in pre-training from scratch for future work.
>
> It is worth noting that SLM not only achieved excellent results in continual pre-training for math but also showed significant improvements in general knowledge, such as MMLU (+11.3%), and in coding tasks, such as MBPP (p@10, +7.8%) and HumanEval (p@10, +10.6%), on TinyLlama, which has been pre-trained with 3T tokens, as shown in **Figure 5**.
>
> We will revise the relevant claims and sections according to your comments and change the term "pre-training" to "continual pre-training" where appropriate.
>
> ### **W3. Reference model performance**
>
> We will refine the table of experimental results. The performance of the model trained by SLM is can outperform the reference model.  As the results shown in Table 3, Tinyllama-CT (RM) is the reference model used in all experiments of this table.
>
> ### **W4. Improving SLM by weighting tokens**
>
> Thank you for your suggestions. To guarantee the simplicity and convenience of the experiment, only the selection was carried out in this paper, and rich analysis experiments were conducted for better understanding. Just as you mentioned, we have reserved the weighting method for future work in Appendix B.
>
> ### **Q1a. The contents of each token category**
>
> In Figures 11-14, we have provided examples including four categories of tokens and several examples of selected tokens. Due to the excessive length of the examples, we have placed a small number of visualization results in the appendix for interested readers to observe the actual situation of tokens in different contexts. According to your opinion, we will increase the number of related examples in revision.
>
> ### **Q1b. Token category statistic**
>
> This sounds like an interesting statistic and we consider supplementing it in Appendix G. According to our observation, in most cases, each token will belong to different categories in different contexts, but there will be a tendency of the categories. For example, some token with word suffixes frequently appear in the L→L category.
>
> ### **Q1c. Figure 14**
>
> The intention of placing Figure 14 in the appendix was to demonstrate that, even with the same context during the training process, the scores obtained for each token at different training stages were not completely consistent. This was done to help better understand the training dynamics of SLM.
>
> ### **Q2. Release plan**
>
> We will release the pre-trained and fine-tuned Rho-1B and Rho-7B. Meanwhile, the data and code will be open-sourced after review process.
>
> ### **Q3. Impact of SLM on memorization of pretraining data**
>
> This is a very interesting question! Methods like Selective Language Modeling (SLM), which selectively include or exclude token losses during training, show great potential in addressing issues such as memorization and repetition in LLMs. And we have also discovered that recent work using similar methods has yielded promising results. [1] We believe this is an area worthy of further exploration!
>
> ### **Other suggestions**
>
> Finally, thank you very much for your meticulous suggestions! They are very helpful for us in improving the paper!
>
> ---
>
> [1] Hans, Abhimanyu, et al. "Be like a Goldfish, Don't Memorize! Mitigating Memorization in Generative LLMs." *arXiv preprint arXiv:2406.10209* (2024). https://arxiv.org/pdf/2406.10209

---

> > ### Comment · Reviewer_pLtQ · 2024-08-12
> >
> > Thanks for the detailed response! For W1, I do think it would be interesting to present a FLOPs comparison of SLM and base pretraining (e.g. measure the FLOPs required for training the reference model + scoring), but it's not crucial to the narrative of the paper. I agree that this scoring pass could be seen as another type of (complex) data preprocessing, and thus not really a factor counted in training-time cost.
> >
> > I think this is a great paper, and I've raised my score 6->7.

---

### Official Review · Reviewer_3eTg · 2024-07-05

**Soundness:** 3
**Presentation:** 3
**Contribution:** 3
**Rating:** 7
**Confidence:** 4

**Summary:**

The authors explore how loss for specific tokens changes in continued pre-training and note that they fall into four categories (high->high, high->low, low->high, low->low) with each category having at least 10%. They run continued pre-training on tokens that are learnable and domain-useful (judged by reference model) and find that this leads to higher accuracy with less tokens used. The main results are in the math domain, but there are also a variety of other results (tool use generalization, general domain, etc.).

**Strengths:**

For originality, I'm not deeply acquainted with related work, but it seems that the authors are (based on Related Work section in appendix). This works seems novel and well-contextualized with respect to related work. The experiments are of high quality and explore a few domains/problems. The paper is generally clear and easy to read. I think this work seems significant in that future research/application could use it (especially with a particularly low-quality dataset).

**Weaknesses:**

* The greatest weakness (in my opinion) is that "tokens" in many cases could refer to "number of tokens after % filtering" and "total number of tokens before filtering." This may be making some results misleading. This ambiguity is present throughout the paper. Just one example is in 3.3 - is 80B the total before filtering or after filtering?
    * Relatedly, in the case when it's after the % filtering, the "x-axis" should be total number of tokens before filtering in my opinion, because the % filtering isn't making training cheaper. I believe the results will still look good after these changes (the numbers in Table 1, for example, are great). But I think Figure 1, for example, should use total tokens (not after % filtering) if it's not already.
* It seems like OpenWebMath is very messy. How would this method work on a clean dataset (like the small, high-quality reference dataset). Is the benefit of the method mostly in "cleaning" the data, or in selecting useful tokens?
* How was the % for filtering chosen for the experiments?

**Questions:**

Please see weaknesses above for some explicit and implicit questions.

**Limitations:**

Yes

---

> ### Author Rebuttal · Authors · 2024-08-07
>
> Dear Reviewer 3eTg,
>
> Thank you for your thoughtful review and for recognizing the novelty, effectiveness, and significance of our work!
>
> ### **Definition of “number of tokens”**
>
> > The greatest weakness (in my opinion) is that "tokens" in many cases could refer to "number of tokens after % filtering" and "total number of tokens before filtering." This may be making some results misleading. This ambiguity is present throughout the paper. Just one example is in 3.3 - is 80B the total before filtering or after filtering?
>
> > Relatedly, in the case when it's after the % filtering, the "x-axis" should be total number of tokens before filtering in my opinion, because the % filtering isn't making training cheaper. I believe the results will still look good after these changes (the numbers in Table 1, for example, are great). But I think Figure 1, for example, should use total tokens (not after % filtering) if it's not already.
>
> For a fair comparison, we default to using “number of tokens **before** filtering” when we say “tokens” throughout the paper, just as you suggested.
>
> On this basis, when referring to the actual selected tokens used for training **after** filtering, we usually make a special note, such as in L165:
>
> > "15 billion tokens (selecting 10.5 billion tokens)".
>
> It is worth noting that in Tables 1, 3, and 5, the total number of input tokens (i.e., before filtering) for all comparative experiments is the same. Therefore, to facilitate the presentation of the actual training tokens, we use “Uniq. Toks” to denote unique tokens and “Train Toks” to denote selected tokens. This is additionally clarified in the caption of Table 1.
>
> We appreciate your suggestion to define "number of tokens" more clearly, and we will specifically clarify this point in the paper and table captions to make it more explicit.
>
> ### **OpenWebMath**
>
> > It seems like OpenWebMath is very messy. How would this method work on a clean dataset (like the small, high-quality reference dataset). Is the benefit of the method mostly in "cleaning" the data, or in selecting useful tokens?
>
> As we know, OpenWebMath is currently a relatively high-quality open-source pre-trained dataset for mathematics. It has undergone meticulous cleaning and is widely adopted by various models [1]. Nevertheless, scoring with the reference model can still identify "messy" and dirty data, as shown in Appendix C, particularly the noisy tokens within a single document that cannot be filtered by traditional document-level methods.
>
> Secondly, we believe that the concepts of "cleaning" the data and "selecting useful tokens" overlap. SLM achieves the effect of filtering the data by selecting useful tokens for the relevant field from the pre-trained data. Therefore, we assert that both "cleaning" and "selecting" are present in our settings.
>
> Moreover, our work focuses on pre-training. Whether the SLM method can still be effective with high-quality data, such as in SFT, remains a question for future work. We believe this is a good scenario to validate whether SLM can bring benefits solely through "selecting”.
>
> ###  **How to choose k% for filtering**
>
> > How was the % for filtering chosen for the experiments?
>
> In the pre-experiment phase, we trained on small-scale data subsets with different percentages and observed the model's accuracy to guide our selection, as shown in **Figure 9**. We found that within a certain percentage range (e.g., from 50% to 70%), there was no significant difference in performance. Therefore, we directly chose 60% and 70% for the larger-scale experiments.
>
> ---
> [1] Azerbayev, Zhangir, et al. "Llemma: An open language model for mathematics." *arXiv preprint arXiv:2310.10631* (2023). https://arxiv.org/pdf/2310.10631

---

### Official Review · Reviewer_QeQV · 2024-07-13

**Soundness:** 3
**Presentation:** 4
**Contribution:** 4
**Rating:** 9
**Confidence:** 4

**Summary:**

The authors propose a method to train LLMs on the most influential tokens selectively. They suggest training a reference model on a small high-quality corpus using the standard CLM loss. They then compute the excess loss of each token in the training corpus as a difference in losses of the reference model and target model on that token. Finally, the target model is trained of the k% subset of the training corpus with the highest excess loss. The paper describes continued pre-training experiments for 1b and 7b models to demonstrate the effectiveness of this method. The experiments show improvements compared to standard continually pre-trained baselines and some open models in terms of performance on popular benchmarks and training efficiency (number of training tokens required to match the performance of open models).

**Strengths:**

The Selective Language Modelling method proposed in the paper is a novel approach to pre-training LLMs. The authors' experiments demonstrate significant improvements in training efficiency which is an important problem in LLM pre-training. The paper also describes a study of LLM training dynamics which could provide useful insights to other researchers working in the field for further exploring efficient token selection strategies for LLM pre-training.

**Weaknesses:**

The experiments in the paper are performed in the continued pre-training setting and the impact of the original pertaining performance is not discussed in the paper. It is possible that the method might not work well if the base model is undertrained.

**Questions:**

The end of section 2 talks about how tokens are selected for training in practice, it says that token selection can be implemented by ranking the tokens by their excess losses and only using the top k% for training. This seems like a crucial detail to ensure that the efficiency gains translate to training wall-clock time. How can this be done while maintaining token sequencing within samples?

**Limitations:**

While the authors have discussed the limitations of their work, an additional direction for the future could be to study the use of multiple domain specialist reference models to select influential training tokens.

---

> ### Author Rebuttal · Authors · 2024-08-07
>
> Dear reviewer QeQV,
>
> Thank you for your comprehensive review and positive remarks!
>
> ### **Pre-training from scratch**
>
> > The experiments in the paper are performed in the continued pre-training setting and the impact of the original pertaining performance is not discussed in the paper. It is possible that the method might not work well if the base model is undertrained.
>
> Due to budget constraint, we conducted experiments on a continued pre-training setting to verify the effectiveness of the SLM. We will rephrase the description in the paper to make it more rigorous and leave the original pre-training setting as future work.
>
> ### **Detailed implementation of SLM**
>
> > The end of section 2 talks about how tokens are selected for training in practice, it says that token selection can be implemented by ranking the tokens by their excess losses and only using the top k% for training. This seems like a crucial detail to ensure that the efficiency gains translate to training wall-clock time. How can this be done while maintaining token sequencing within samples?
>
> In the implementation, we maintained the token sequencing in each sample and simply removed the losses of tokens with lower scores in the output part (therefore, the forward computational cost of token loss was not reduced). Due to the removal of these low-score tokens, the SLM achieved better performance compared to training with all tokens when using the same amount of input tokens, thereby improving data efficiency.

---

> > ### Comment · Reviewer_QeQV · 2024-08-13
> >
> > Thank you for the clarification! I will retain my scores.

---

### Public Comment · ~Andreas_Kirsch1 · 2024-12-12
**Important Prior Work Attribution: RhoLoss and Rho-1**

I am writing to note an important oversight in attribution that should be addressed, particularly given this paper's selection as a runner-up for the [NeurIPS 2024 Best Paper Award](https://blog.neurips.cc/2024/12/10/announcing-the-neurips-2024-best-paper-awards/)---a recognition that highlights its visibility.

The paper presents "Selective Language Modeling (SLM)" which uses an "excess loss" scoring mechanism that appears to be fundamentally equivalent to the RhoLoss method introduced in "Prioritized Training on Points that are Learnable, Worth Learning, and Not Yet Learnt" ([Mindermann et al., ICML 2022](https://arxiv.org/abs/2206.07137)). While the Mindermann et al. paper is cited, it is only briefly mentioned as related work in sample selection, without acknowledging or comparing against the fact that it introduces the same core method.

The similarity extends beyond just the technical approach:

1. The core scoring mechanism comparing losses between a reference model and training model
2. The name "Rho-1" itself, which is remarkably similar to "RhoLoss"
3. The underlying principle of identifying valuable training examples through loss comparisons

While the current work makes valuable contributions in applying and extending these ideas to token-level selection in language model pre-training, the original RhoLoss work should be properly acknowledged and discussed. This is particularly important given that the authors were made aware of this connection [in April 2024 (on HuggingFace) and agreed](https://web.archive.org/web/20240618180512/https://huggingface.co/papers/2404.07965) to address it in future revisions. However, the current camera-ready version still lacks this critical attribution and comparison.

The authors' contributions in extending these ideas to token-level selection and demonstrating their effectiveness in large-scale language model training are significant and worthy of recognition. However, proper attribution of the foundational ideas and a direct comparison to RhoLoss would strengthen the paper and better position it within the broader research context.

I recommend adding a clear discussion of RhoLoss in the related work section, explicitly acknowledging how the current work builds upon and extends these ideas, and including a direct comparison of the methods to highlight both the similarities and the novel contributions in token-level application.

---

> ### Public Comment · ~Zhenghao_Lin1 · 2024-12-12
>
> Thanks for your comments. In response to the concerns you raised, we provide the following replies, and we hope that our responses will address your concerns:
>
> 1. Our work takes a token-level perspective, while your work focuses on sample-level data selection, which is not the primary focus of our research. Meanwhile, we concentrate mainly on the pre-training process, investigating the importance of tokens during this phase, as reflected in our title, "Not All Tokens Are What You Need for Pretraining."
>
> 2. The name "rho-1" follows our existing "phi-1/phi-3.5" series (and we are also developing "sigma-1"), signifying "information density." This is purely coincidental with your 2022 work. We primarily focused on recent works related to data selection in the pre-training of large language models, which led to an oversight of your work and, consequently, some misunderstandings. At your previous request, we have added a reference to "rho-loss" as an "online batch selection method" in the related work section of the camera-ready version. We would like to adding more discussion about your work in our next version.
>
> 3. The method for selecting tokens is flexible. For instance, we provide alternative approaches for selecting tokens in Appendix H.  From our design perspective, we initially trained a reference model on high-quality data, aiming to model a high-quality distribution. We then directly scored and filtered the pre-training tokens using this model. However, we found that this approach tends to select overly simple tokens and fails to reflect the training dynamics of the target model. To address this, we incorporated the current model’s loss to calculate an "excess loss." We discovered that this method effectively selects clean tokens with appropriate difficulty for learning, preventing the model from discarding "challenging tokens" and resulting in better performance.

---

> ### Public Comment · ~Andreas_Kirsch1 · 2024-12-13
> **Re: RhoLoss Attribution and Implementation Efficiency Concerns**
>
> Thank you for your reply. I appreciate the additional context, but I must respectfully disagree with several points:
>
> 1. While your application to token-level selection is indeed novel and valuable, the core method - computing excess loss between a reference and training model - is identical to RhoLoss, the reducible hold-out loss. That you independently arrived at the same solution actually strengthens the case that this is a fundamental approach worth acknowledging properly. Your discovery that it works well at the token level is a significant contribution, but it builds on the same mathematical foundation.
>
> 2. Regarding the naming: While I understand the phi/rho/sigma series explanation, it's worth noting that when you were made aware of this similarity in April 2024, you had the opportunity to either change the name or explicitly acknowledge and explain this coincidence. The current minimal citation of Mindermann et al (2022) as just another "online batch selection method" doesn't adequately address this.
>
> 3. Your explanation of the method's development is interesting, but it actually highlights how similar the underlying principles are. RhoLoss was also motivated by the need to balance between selecting learnable examples while avoiding either too simple or too noisy ones, as you can see in the extensive treatment of this in the paper. This is why we used the difference between reference and training model losses - exactly the same solution you arrived at two years later.
>
> ---
>
> To illustrate just how fundamentally similar the methods are, here is the pseudocode for both approaches:
>
> RhoLoss (Mindermann et al., ICML 2022):
> ```python
> def rholoss(batch, model, reference_model):
>     # losses shape (batch_size,) for classification
>     train_losses = compute_losses(batch, model)
>     ref_losses = compute_losses(batch, reference_model)
>     excess_loss = train_losses - ref_losses
>     threshold = np.percentile(excess_loss, 100 * (1 - k_percent))
>     mask = excess_loss > threshold
>     train_model(batch[mask], model)
> ```
>
> Selective Language Modeling (this work, 2024):
> ```python
> def selective_language_modeling(batch, model, reference_model):
>     # losses shape (batch_size, seq_len) for lm
>     train_losses = compute_losses(batch, model)
>     ref_losses = compute_losses(batch, reference_model)
>     excess_loss = train_losses - ref_losses
>     threshold = np.percentile(excess_loss.flatten(), 100 * (1 - k_percent))  # Only difference: .flatten()
>     mask = excess_loss > threshold
>     train_model(batch[mask], model) # backprop through train_losses[mask]
> ```
>
> **Note**: The _only_ difference is the `.flatten()` operation to go from `(batch_size, seq_len)` to `(batch_size*seq_len)` for token-wise selection. The core idea of computing excess loss and sub-selection is identical.
>
> ---
>
> Additionally, after spending more time with the paper, I have concerns about the efficiency claims. While you demonstrate faster convergence in terms of number of training steps, the actual wall-clock time savings deserve more rigorous analysis. Due to the nature of transformer attention:
>
> - All tokens in a sequence must be processed in the forward pass to maintain context
> - Attention must be computed across all tokens in the sequence
> - Hence, if any token in a sequence is selected, backpropagation will still have to be computed for all prior tokens in that sequence (due to causal attention).
>
> This means that while the average loss might be computed for the sub-selected tokens, the actual computational savings would be much smaller than implied by the "10x faster" and "5x faster" claims in your paper since each step still requires nearly the same computations for backpropagation as standard training.
>
> ---
>
> Would you be willing to update the paper to include:
>
> 1. A clear acknowledgment of RhoLoss (Mindermann et al., ICML 2022) as prior work that developed the core method of excess loss computation
> 2. A proper comparison and discussion in the related work section, highlighting both the similarities and your novel contributions in token-level application (also e.g. in regards to _your analysis of token losses as L-L, H-L, H-L, H-H vs Mindermann et al's analysis of samples as being noisy, redundant, not relevant, or worth training on_)
> 3. A more detailed analysis of actual wall-clock time savings, including the computational overhead of computing both reference and training model losses, and the limitations imposed by transformer architectures.
>
> This would strengthen your paper while maintaining appropriate scientific attribution and providing a more complete analysis of the method's efficiency.
>
> _PS: Regarding the last point, it seems Reviewer __pLtQ__ had similar concerns about the actual training speed vs. data efficiency, but the revision you promised in your response did not seem to have happened. It would seem a good idea to fulfill promises made during the response period._

---

### Public Comment · ~Ofek_Levy1 · 2025-01-15
**Please add a full explanation of Figure 7 in the ablation study**

Hello, I read your article in detail and found it very interesting.
In figure 7, in the ablation study, you ask how does the selected/un-selected tokens' loss correlate with downstream performance.
The graphs you provide seem to align with the intuition that a decrease in the selected tokens' loss correlates with higher downstream performance, and the opposite for the un-selected tokens.

Unfortunately you do not provide an explanation for how this data was generated.
Particularly I'm not sure I see what differs between different points of the same color on the graphs.
If I understand correctly, every row of five points (five colors) represents the downstream performance of a model whose selected (/un-selected) tokens' loss, at each of the five checkpoints, is the x-value.
So, what changes between different rows of points? Are these different models, later epochs, or something else?

I believe this to be a crucial diagram in your study, and would like to understand it properly.

Thank you!

---

### Decision · Program_Chairs · 2024-09-25

**Decision:**

Accept (oral)

**Comment:**

This paper presents a very interesting method for filtering pre-training data for LLM training. The idea is basically first to create a golden data set for which we are certain of the quality of the tokens (no noisy tokens etc.), train a language model on that, and then use that language model to assign a score for each token in a large target pre-training corpus which we want to train the final LLM on. Then the scores per token are used to predict whether or not the loss from that token should be used in training the target language model. This ensures that we are only using the highest quality data aligned with the golden corpus in training the target LLM.

The paper is clearly written with a solid set of experiments and analysis.